# Efficient Prediction of Large Protein Complexes via Subunit-Guided Hierarchical Refinement

**Chixiang Lu**[1], **Yunhua Zhong**[1], **Shikang Liang**[2], **Xiaojuan Qi**[3], **Haibo Jiang**[1]

[1]Department of Chemistry, The University of Hong Kong
[2]School of Biomedical Sciences, The University of Hong Kong
[3]Department of Electrical and Electronic Engineering, The University of Hong Kong
`u3590540@connect.hku.hk; xjqi@eee.hku.hk; hbjiang@hku.hk`

## Abstract

State-of-the-art protein structure predictors have revolutionized structural biology, yet quadratic memory growth with token length makes end-to-end inference impractical for large complexes beyond a few thousand tokens. We introduce Hier-AFold, a hierarchical pipeline that exploits the modularity of large complexes via PAE-guided (Predicted Aligned Error) subunit decomposition, targeted interface-aware refinement, and confidence-weighted assembly. PAE maps localize rigid intra-chain segments and sparse inter-chain interfaces, enabling joint refinement of likely interacting subunits to capture multi-body cooperativity without increasing memory. HierAFold matches AlphaFold3 accuracy, raises success rates from 49.9% (CombFold) to 73.1% on recent PDB set. While for large complexes, it cuts peak memory by $\sim 25\,\mathrm{GB}$ on a 4,000 token target ($\sim 40\%$), successfully models complexes with over 5,000 tokens that are Out-Of-Memory for AlphaFold3, and raises success rates by two-fold compared with CombFold. The code is available at https://github.com/Luchixiang/HierAFold.

## 1 Introduction

The advent of deep learning in protein folding prediction has transformed structural biology (Du et al., 2021; Senior et al., 2020; Jumper et al., 2021). Models such as AlphaFold2 (Jumper et al., 2021; Evans et al., 2021) and RoseTTAFold (Baek et al., 2021) and their successors AlphaFold3 (Abramson et al., 2024) and RoseTTAFold All-Atom (Krishna et al., 2024) now achieve unprecedented accuracy in modeling biomolecular interactions, including proteins, nucleic acids, and small molecules. Despite these advances, scaling to very large complexes, typically on the order of thousands of tokens, remains a major bottleneck. Memory usage in transformer-like modules (notably triangle updates and attention) grows quadratically with token length, making end-to-end inference prohibitive beyond $\sim$4–5k tokens on modern GPUs (for example, a 4,500-token complex can require 80 GB of GPU memory) (Abramson et al., 2024).

Current approaches attempt to mitigate these limits by by partitioning large tensors (Ahdritz et al., 2024), or by predicting pair/triple complexes and relying on Monte Carlo Tree Search (MCTS) (Chaslot, 2010; Bryant et al., 2022; Chim & Elofsson, 2024) or combinatorial assembly (Shor & Schneidman-Duhovny, 2024) for global assembly. Consequently, these methods remain insufficiently scalable or miss crucial multi-body cooperativity, limiting the accuracy of the complexes with higher-order coupling (Peng et al., 2023).

Large complexes (e.g., chaperonins, synthetic multi-enzyme circuits) are inherently modular, a property we exploit for efficient prediction: (i) each peptide chain comprises one or more distinct folded subunits and intrinsically disordered regions; and (ii) inter-chain interaction interfaces critical for assembly are typically restricted to specific subunits and regions, and sparse relative to total residues (Reichmann et al., 2005). This suggests that end-to-end prediction is unnecessary: one can first predict interacting interfaces between chains and then assemble chain-level structures accordingly.

To make this strategy practical, we need an automatic way to detect subunits and interfaces. Predicted Aligned Error (PAE) quantifies the expected positional error of residue $i$ after aligning on residue $j$ and correlates strongly with rigid-body coherence within domains and the reliability of inter-chain interfaces (Jumper et al., 2021). Thus, we explore using PAE to drive automatic modular decomposition and interface detection. We find that coarse predictions yield PAE matrices that naturally partition into low-variance diagonal blocks (coherent subunits) and localized low PAE off-diagonal patches (interaction interfaces) (Fig. 1).

Motivated by these insights, we introduce HIERAFOLD (Fig. 2), which exploits modularity to assemble peptide chains in a hierarchical, coarse-to-fine manner, drastically reducing memory for large protein complex prediction. **(i) Coarse stage**. HIERAFOLD generates efficient pairwise predictions, producing PAE matrices and initial 3D structures for all chain pairs using a distilled, few-step diffusion model (Song et al., 2023). From the PAE matrices, it segments each chain into structurally coherent subunits (dark diagonal blocks) and identifies candidate interface subunits (dark off-diagonal patches) across chains. **(ii) Fine stage**. For each chain, HIERAFOLD jointly refines its structure with candidate interface subunits from other chains, capturing multi-body cooperativity and yielding higher fidelity models while keeping memory bounded. **(iii) Assembly**. Finally, HIERAFOLD merges the refined partial structures

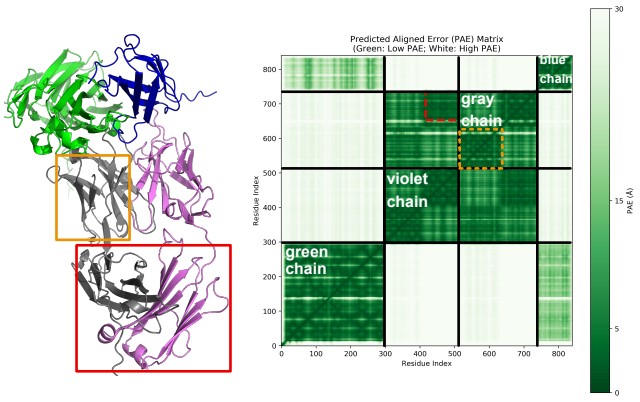

Figure 1: Example of dividing subunits along a chain and identifying the inter-chain interface based on the PAE matrix. Each color denotes a peptide chain. The black lines in PAE are chain borders. The orange square marks a divided subunit in the gray chain. The red rectangle highlights the low-PAE region at an inter-chain interacting subunit, which is sparse relative to the total residues.

via a confidence-aware alignment module that weights alignment by estimated confidence, prioritizing well-predicted residues. By using sparse subunits as cross-chain bridges and explicitly modeling higher-order cooperativity among multiple subunits without inflating memory, HIERAFOLD overcomes key limitations of prior methods.

HIERAFOLD achieves accuracy on par with AlphaFold3 for protein–protein and protein–ligand benchmarks, while cutting GPU memory use by about 40% on large token counts. Unlike baselines that fail with Out-Of-Memory errors, it successfully models complexes exceeding 5,000 tokens, delivering a two-fold improvement in success rate compared to CombFold.

In summary, our contributions include:

- **HIERAFOLD:** a hierarchical, coarse-to-fine pipeline for large protein complexes that keeps memory bounded while scaling beyond conventional token limits.

- **PAE-guided modular decomposition:** automatically segments chains into rigid subunits and pinpoints localized inter-chain interfaces, removing the need for expert curation.

- **Interface-aware refinement and assembly:** jointly refines each chain with its most likely partner subunits to capture multi-body cooperativity, and aligns chains via subunit interfaces using a confidence-aware module for greater robustness.

- **Strong empirical performance:** HIERAFOLD matches a reproduced AlphaFold3 baseline on the benchmark, successfully models complexes exceeding 5,000 tokens, and outperforms other methods by a large margin.

## 2 RELATED WORK

**AlphaFold.** Protein structure prediction has progressed from physics-based simulations to data-driven deep learning, with AlphaFold2 (Jumper et al., 2021) establishing a landmark by leveraging multiple sequence alignments (MSAs) and equivariant attention networks to achieve near experimental accuracy. AlphaFold3 (Abramson et al., 2024) extends this paradigm to multi-chain protein complexes, nucleic acids, and small molecular ligands through a unified diffusion-based framework (Ho et al., 2020).

**Tokens in AlphaFold3.** To manage memory, AlphaFold3 represents inputs primarily at the token level instead of the atom level. A single token corresponds to one amino acid residue, one nucleotide, or one atom of a ligand/non-standard residue (Abramson et al., 2024).

**Large Complex Prediction.** The Pairformer module in AlphaFold introduces *quadratic memory growth* in token number, making end-to-end inference impractical for large complexes. To mitigate memory limits, OpenFold (Ahdritz et al., 2024) splits large tensors and processes them piecewise to trade throughput for lower memory, but the pair representation and attention still dominate memory and scale poorly. Beyond such tensor-level tricks, MoLPC (Bryant et al., 2022) predicts pair/triple subunits and uses Monte Carlo Tree Search (MCTS) (Chaslot, 2010) for global assembly, which underperforms on heteromeric complexes due to reliance on accurate symmetry and stoichiometry (Shor & Schneidman-Duhovny, 2024; Chim & Elofsson, 2024). Similarly, CombFold (Shor & Schneidman-Duhovny, 2024) assembles large complexes via combinatorial and hierarchical assembly, but depends on high-quality predefined subunits, often requiring expert curation and yielding incomplete or biased models. Because these pipelines emphasize pairwise interactions, they also miss crucial multi-body cooperativity, effects from multiple interacting chains, limiting accuracy on complexes with higher order coupling (Peng et al., 2023).

Meanwhile, docking-based approaches that rely on pairwise protein–protein docking (Inbar et al., 2003; Esquivel-Rodríguez et al., 2012; Kuzu et al., 2014) are more memory-friendly but lag behind deep learning methods in accuracy. Consequently, current approaches remain insufficiently scalable or accurate, overly specialized, or insufficiently automated for modeling large complexes.

**Protein Subunit/Domain Segmentation.** Protein domain segmentation has been studied, with two primary categories of methods: (i) *Domain annotation databases*, such as CATH (Sillitoe et al., 2021) and ECOD (Schaeffer et al., 2017), which provide curated evolutionary classifications and domain boundaries based on sequence and structure similarity. (ii) *Computational prediction methods,* including sequence-based predictor (Zhu et al., 2023), structure-based predictor (Zhang et al., 2023; Yu et al., 2022), deep-learning approaches (Lau et al., 2023; Wells et al., 2024). (iii) *Large-scale efforts,* such TED, which annotate structural domains across the protein universe by integrating these advanced tools (Lau et al., 2024). However, these approaches require additional inference-time computation. What's more, they are optimized for evolutionary domain definitions and do not identify cross-chain interface subunits.

## 3 PRELIMINARIES

**Confidence matrices.** AlphaFold family models output complementary confidence estimates to assess prediction reliability: (i) *pLDDT* (predicted Local Distance Difference Test) provides per-residue accuracy estimates (Mariani et al., 2013); (ii) *pTM* (predicted TM-score) approximates the expected TM-score to the unknown native structure, measuring overall fold accuracy; (iii) *ipTM* (interface predicted TM-score) quantifies the relative placement of chains or interacting components. In AlphaFold3, a weighted combination of ipTM, pTM, and clash penalties is used to rank multiple samples.

**Predicted Aligned Error (PAE).** The Predicted Aligned Error (PAE) matrix from AlphaFold estimates the positional error (in Ångströms) between any two residues after aligning the predicted and true structures on one of them. Low PAE values indicate high confidence in the relative positions of residues, suggesting a structurally well-defined region, while high values imply uncertainty due to either protein flexibility or a poorly predicted region.

Importantly, PAE provides structural cues beyond per-residue scores since it encodes the model's belief about the *relative flexibility and coherence of structural regions*. Intuitively, when two re-

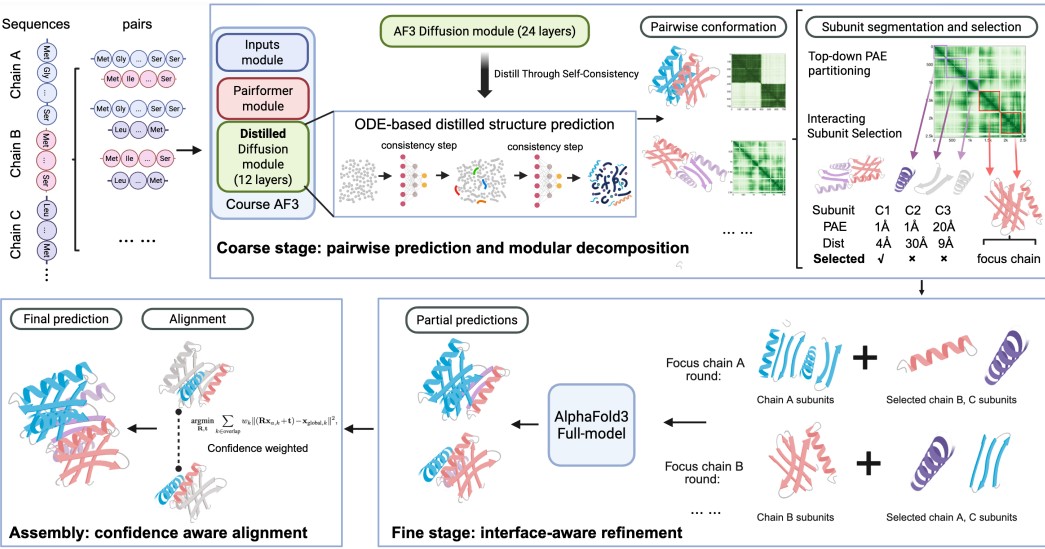

Figure 2: An overview of HIERAFOLD. Given the input sequences of peptide chains, the coarse stage generates pairwise conformation of chains with predicted PAE using a distilled diffusion module. Using the PAE matrices, it divides each chain into structurally coherent subunits (dark diagonal blocks) and detects potential interface subunits (dark off-diagonal patches) across chains. Structure of the focus chain with the selected subunits from other chains is refined using AlphaFold3 full model. Multiple partial predictions are aligned and assembled by estimating the transformation matrix with a confidence-weighted method.

gions of a protein move coherently as a rigid unit, the model can predict their relative positions with high confidence, leading to low PAE. Conversely, residues belonging to different domains or flexible linkers exhibit higher relative uncertainty and therefore higher PAE, providing a meaningful structural signal for segmentation. As illustrated in Fig. 1: (i) *low-value diagonal blocks* often align with intrinsic, independently folded subunits within a chain, which represent functional domains rather than individual residues. Biologically, these subunits, instead of individual amino acids, represent functional modules and act as units to facilitate interaction between peptide chains (Pawson, 1995). (ii) *off-diagonal low-value blocks* correspond to confident inter-chain or inter-subunit contacts, highlighting potential interfaces critical for assembly. This observation suggests that PAE can be leveraged to heuristically segment large complexes into subunits and to identify interacting interfaces, enabling modular decomposition into tractable parts for scalable structure prediction.

## 4 HIERAFOLD

For a large protein complex with peptide chains $\{C_1, C_2, \ldots, C_M\}$ with a total of $N$ tokens, end-to-end prediction quickly becomes computationally infeasible as $N$ grows. HIERAFOLD overcomes this by reformulating the prediction problem as a series of smaller, computationally tractable refinement tasks, guided by the protein complex's inherent modularity. Instead of processing the entire complex at once, it iteratively predicts the structure of a designated focus chain $C_a$ in the context of only its most relevant interacting subunits, denoted as $\mathcal{N}(C_a)$. After all chains are refined in the corresponding localized context, the results are integrated into a complete structure through a confidence-aware assembly step.

The full pipeline (Fig. 2) proceeds in three stages: (i) Coarse stage: generate fast pairwise predictions and derive modular decomposition from PAE matrices; (ii) Fine stage: perform high resolution, interface-aware refinement of each focus chain $C_a$ together with its selected subunits $\mathcal{N}(C_a)$ from other chains; and (iii) Assembly stage: merge partial structures using a confidence-weighted algorithm. This hierarchical design captures both local structural accuracy and long-range cooperative interactions, while keeping memory demands within practical limits.

## 4.1 COARSE STAGE: PAIRWISE PREDICTION AND MODULAR DECOMPOSITION

The first stage aims to decompose a whole complex into several tractable parts based on the inherent modularity of the complex. This process consists of three steps: fast pairwise prediction, PAE-guided subunit segmentation, and interacting subunit selection.

**Fast Pairwise Conformation Prediction** To obtain the initial structural and PAE matrix for modular decomposition, given the peptide chain sequences, the first step of HIERFOLD is to perform pairwise structure prediction on every pair of chains $(C_i, C_j)$. However, running a full AlphaFold3-style inference with hundreds of diffusion steps for all $\binom{M}{2}$ pairs would be prohibitively slow. To accelerate this, we employ a consistency-distilled structural predictor (Song et al., 2023; Song & Dhariwal, 2023). The goal is to train a model $f_\theta$ that can generate coordinates $\mathbf{x}_0$ of the chain pair $(C_i, C_j)$ from a random noise input $\mathbf{x}_T$ in a single step or few steps: $\mathbf{x}_0 = f_\theta(\mathbf{x}_T, T)$.

We train this model using a consistency objective, which enforces that the outputs of the model at different points along the probability flow ODE's trajectory are self-consistent (Song et al., 2020b; 2023). Let $\theta^-$ be the parameters of a target model (an exponential moving average of the model's parameters $\theta$). The loss is defined on pairs of adjacent time steps $(t_n, t_{n+1})$: Given noisy coordinates $\mathbf{x}_{t_{n+1}}$ and an ODE-based backstep estimate (Song et al., 2020a) $\hat{\mathbf{x}}_{t_n}$:

$$L_{\text{con}} = \mathbb{E}\left[d(f_\theta(\mathbf{x}_{t_{n+1}}, t_{n+1}, c), f_{\theta^-}(\hat{\mathbf{x}}_{t_n}, t_n, c))\right], \tag{1}$$

where $c$ represents the conditioning feature extracted from the Pairformer. $\hat{\mathbf{x}}_{t_n}$ is the one-step ODE solution starting from $\mathbf{x}_{t_{n+1}}$, and $d$ is a distance metric in coordinate space (e.g., a clamped L2 distance). To further reduce computational cost, the distilled model uses a shallower Pairformer (12 vs. 24 blocks). This process yields a set of coarse coordinate predictions $\{\mathbf{X}_{ij}\}$ and corresponding PAE matrices $\{\mathbf{P}_{ij}\}$ for all chain pairs, which serve as the input for the subsequent stages.

**PAE-Guided Subunit Segmentation for Biological Modularity** With the predicted pairwise PAE matrix, the next step is to segment each chain into the intrinsically independent folded subunits. For a chain $C_i$ of length $L$, its intra-chain PAE matrix $\mathbf{P}_{ii} = \mathbf{P}_{ij}[: L, : L]$ reveals its internal structural confidence. We observe that diagonal blocks of PAE matrices with low average PAE values correspond to compact domains or stable structural motifs (e.g., alpha-helices and beta-sheets), as shown in Fig. 1.

To achieve this automatic subunit segmentation, we employ a recursive, top-down partitioning algorithm that iteratively identifies the most probable boundaries within a protein segment. For any given segment defined by residue indices $[i, j]$, the method searches for the optimal split point $k$ (where $i < k < j$) that maximizes the uncertainty between the two resulting sub-segments, $S_1 = [i, k]$ and $S_2 = [k, j]$. This uncertainty is quantified by the mean inter-subunit PAE, $\mathcal{P}_{\text{inter}}(k)$, calculated as the average PAE over the off-diagonal blocks of $\mathbf{P}$:

$$\mathcal{P}_{\text{inter}}(k) = \frac{1}{2}\left(\frac{\sum_{u=i}^{k-1}\sum_{v=k}^{j-1}\mathbf{P}_{uv}}{(k-i)(j-k)} + \frac{\sum_{u=k}^{j-1}\sum_{v=i}^{k-1}\mathbf{P}_{uv}}{(j-k)(k-i)}\right). \tag{2}$$

The algorithm proceeds as follows: A queue is initialized with the entire chain segment $(0, L)$. In each step, a segment $(i, j)$ is dequeued. We then identify the optimal split point $k^*$ that maximizes $\mathcal{P}_{\text{inter}}(k)$. The split is acceptable if $\mathcal{P}_{\text{inter}}(k)$ exceeds a predefined split threshold, $\tau_{\text{split}}$, and both new sub-segments are larger than a minimum size $L_{\text{min}}$. The new split segments are now queued. This process continues until no more splits can be made, yielding a set of subunits $U_{i,1}, U_{i,2}, \ldots$ for each chain $C_i$.

**Interacting Subunit Selection** Finally, for each "focus chain" $C_a$, we identify a neighborhood of relevant interacting subunits $\mathcal{N}(C_a)$, from all other chains $C_b, b \neq a$, and ignoring other subunits. For subunit $U_{b,j}$ on other chain $C_b$, it is selected if it meets two criteria based on the coarse predictions: high confidence (low PAE) and spatial proximity, which indicates $U_{b,j}$ may be at the interaction interfaces of $C_b$ and focus chain $C_a$. Specifically , for each subunit $U_{b,j}$ on other chain $C_b$, we compute: (i) Mean Interface PAE: $\bar{P}(C_a, U_{b,j})$, the average PAE value between residues in $C_a$ and residues in $U_{b,j}$ from the pairwise matrix $\mathbf{P}_{ab}$. A low value indicates a high probability of interacting. A subunit $U_{b,j}$ is included in the neighborhood set $\mathcal{N}(C_a)$ if $\bar{P}(C_a, U_{b,j}) < \tau_p$. (ii) Minimum Distance: $d_{\min}(C_a, U_{b,j})$, the minimum distance between any center atom in $C_a$ and any

center atom in $U_{b,j}$ from the coarse predicted structure $\mathbf{X}_{ab}$. $U_{b,j}$ is included if $d_{\min}(C_a, U_{b,j}) < \tau_d$. This step effectively identifies sparse cross-chain "bridges" for the refinement stage.

## 4.2 FINE STAGE: INTERFACE-AWARE REFINEMENT

With the pruned context defined for each focus chain, HIERAFOLD performs high-resolution structure prediction. For each focus chain $C_a \in \{C_1, \ldots, C_M\}$, we execute a full AlphaFold3-style inference on the composite input formed by $C_a$ and its selected interacting subunits $\mathcal{N}(C_a)$. Because the input for each run is limited to one full chain plus only the relevant interface subunits from other chains, the total token count remains well below the memory limits of standard GPUs. This step generates a set of $M$ high-resolution, partially overlapping structural predictions, $\{\hat{\mathbf{X}}_1, \hat{\mathbf{X}}_2, \ldots, \hat{\mathbf{X}}_M\}$, each capturing the detailed conformation of a focus chain within its interaction context. However, each partial prediction is in its own coordinate frame.

## 4.3 ASSEMBLY STAGE: CONFIDENCE-AWARE GLOBAL ALIGNMENT

The final stage is to assemble the $M$ partial, well-predicted structures into a single, coherent model of the entire complex. We perform this assembly iteratively. We initialize the global assembly with the highest confidence partial prediction, $\hat{\mathbf{X}}_{\text{best}}$. Then, we iteratively align each remaining partial structure $\hat{\mathbf{X}}_a$ to the current global assembly. To ensure robustness, we use a confidence-weighted Kabsch algorithm (Kabsch, 1976) instead of a simple global superposition. Specifically, we identify the set of atoms that are common to both the global assembly and the partial structure $\hat{\mathbf{X}}_a$. The optimal rotation $\mathbf{R}$ and translation $\mathbf{t}$ are found by minimizing the weighted root-mean-square deviation (RMSD) over the set of atoms common to both structures, where the contribution of each atom pair is weighted by the product of their pLDDT scores:

$$\operatorname*{argmin}_{\mathbf{R},\mathbf{t}} \sum_{k \in \text{overlap}} w_k \|(\mathbf{R}\mathbf{x}_{a,k} + \mathbf{t}) - \mathbf{x}_{\text{global},k}\|^2, \quad \text{where } w_k = \text{pLDDT}(\mathbf{x}_{a,k}) \cdot \text{pLDDT}(\mathbf{x}_{\text{global},k}). \quad (3)$$

This method ensures that the alignment is driven by the most confidently predicted regions of the interface, making the assembly resilient to flexible or poorly predicted loops (low pLDDT). After alignment, the refined coordinates of the focus chain $C_a$ from $\hat{\mathbf{X}}_a$ are merged into the global assembly. For multiple samples, the top-ranked one is always aligned with the top-ranked one. The final structure is ranked using a composite score, similar to AlphaFold3.

## 5 IMPLEMENTATION DETAILS

**Setup and Terminology** Since the official AlphaFold3 training code is not publicly available, all experiments use Protenix (v0.5.0) (Team et al., 2025), an open-source PyTorch reproduction (Paszke et al., 2019), as our baseline and core predictive module. "AlphaFold3" in our experiments refers to predictions made with the pretrained Protenix model. We acknowledge that any performance differences between our results and the original AlphaFold3 paper may arise from implementation or model weight discrepancies.

### 5.1 CONSISTENCY DISTILLATION

To enable efficient pairwise conformation prediction within HIERAFOLD, we trained a consistency-distilled version of the Protenix structure module.

**Architecture** We reduced the computational cost of the structure predictor by halving the number of token-level transformer blocks from 24 to 12. The atom-level transformer stack remained unchanged. All other architectural hyperparameters followed the Protenix v0.5.0 defaults.

**Training** The distilled diffusion model was trained using the Adam optimizer (Kingma, 2014) with a learning rate of $1 \times 10^{-5}$, a 2,000 steps linear warmup, $\beta_1 = 0.9$, $\beta_2 = 0.95$, and a weight decay of $1 \times 10^{-8}$. The noise scheduler is the same as the default in Protenix. Training was performed on the preprocessed complex dataset provided with Protenix, with sequences cropped to a maximum of 512 tokens to fit within GPU memory.

Table 1: Performance comparison on protein-protein and protein-ligand benchmarks. DockQ Success Rate (%) is reported for protein-protein interfaces, and Ligand RMSD $\leq$ 2Å Success Rate (%) is reported for PoseBuster v2 benchmark. "OOM" denotes Out-Of-Memory failure on an 80GB A100 GPU. HIERAFOLD matches the performance of the full AlphaFold3 baseline while successfully predicting large complexes where the baseline fails.

| Method | Recent PDB | | PoseBuster v2 | | Large Complexes (>5k tokens) | |
|---|---|---|---|---|---|---|
| | Oracle | Top-1 | Oracle | Top-1 | Oracle | Top-1 |
| MoLPC | – | 31.3 | – | – | – | – |
| CombFold (w/ AFM) | 49.9 | 48.8 | – | – | 20.9 | 19.9 |
| AlphaFold-Multimer (AFM) | 71.5 | 67.1 | – | – | OOM | OOM |
| CombFold+AF3 | 47.5 | 43.2 | – | – | 20.7 | 19.8 |
| RoseTTAFold All-Atom | – | 55.4 | – | 41.8 | – | – |
| AlphaFold3 (Baseline) | **74.4** | **70.4** | **78.6** | **76.0** | OOM | OOM |
| + HIERAFOLD (Ours) | 72.6 | 68.4 | – | – | 44.1 | 43.6 |
| **HIERAFOLD (Ours)** | 73.1 | 69.0 | 77.4 | 74.7 | **44.5** | **43.9** |

**Sampling / Inference** During the coarse pairwise prediction stage, we run the distilled model for only two iterative refinement steps. For each chain pair, we generate five stochastic samples and select the top-ranked one based on a composite score of ipTM, pTM, and clash penalties. This single, high-ranked coarse prediction is then used for subunit segmentation and selection.

## 5.2 SUBUNIT SEGMENTATION SELECTION POLICY

For subunit For PAE-guided subunit segmentation, the split threshold $\tau_{\text{split}}$ was set to 10.0, and the minimum subunit size $L_{\text{min}}$ was set to 20 residues.

**Selection Policy** The selection of subunits or tokens for the high-resolution refinement stage follows a set of hierarchical rules:

- **Short Chains:** Polymer chains with fewer than 40 tokens are always retained and are not further segmented and selected.

- **Ligands**: For small molecular ligands, all ligand atoms are always retained. All ligands are docked independently with each focus chain and selected subunits from other chains. The final pose is selected based on the highest average atom pLDDT, since we assume the underlying AlphaFold3 scoring is sufficiently discriminative to rank the correct binding partner.

- **Low-Confidence Pairs:** If the coarse prediction for a chain pair $(C_i, C_j)$ yields a maximum ipTM score below 0.2, the interaction is considered unreliable, and the partner chain $C_j$ is entirely excluded from the context of $C_i$ because such pairs are unlikely to represent a true physical interaction.

- **Subunit/Token Filtering:** For polymer chains with 40 or more tokens, we apply the confidence- and proximity-based selection described in Section 3.3. A subunit is selected if its mean interface PAE is less than $\tau_p = 5.0$ or its minimum distance to the focus chain is less than $\tau_d = 20$ Å. Distances are calculated between the center atoms ($C_\alpha$ for amino acids, $C_{1'}$ for nucleotides). If subunit segmentation fails for a given chain, the selection logic reverts to operating at the individual token level.

## 6 EXPERIMENT

### 6.1 EXPERIMENTAL SETUP

**Baseline Model** Again, since the official AlphaFold3 code and weights are not publicly available for training, we use **Protenix (v0.5.0)** as our core predictive module and primary baseline. All mentions

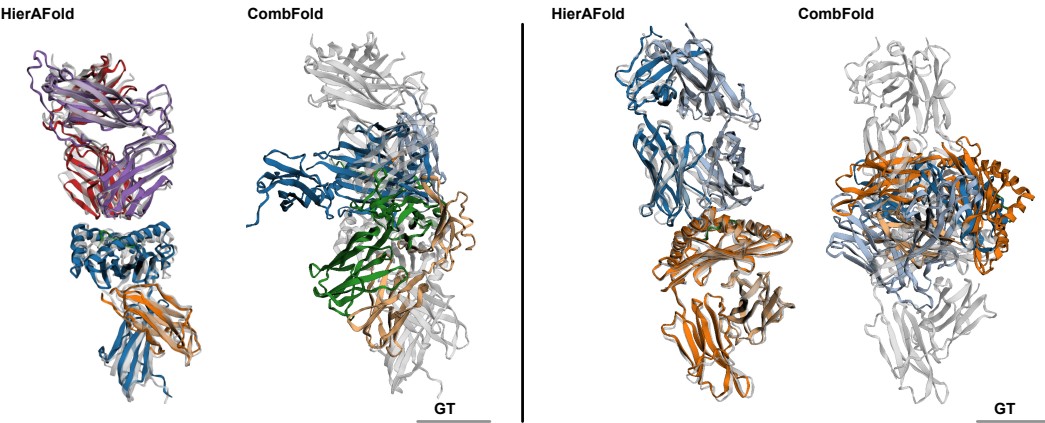

Figure 4: Visualization of complex modeling results comparing HierAFold and CombFold. Left PDB ID is 7na5 and right PDB ID is 8gvi. The gray chains are ground truth. Considering only pairwise interactions causes CombFold to ignore multi-chain interactions, leading to an inaccurate final assembly.

of "AlphaFold3" in our experiments refer to this reproduced version. We use the provided pretrained weights for all predictions.

**Baselines for Comparison** We benchmark HIERAFOLD against several methods: (i)**AlphaFold3 (Protenix)**: The end-to-end Protenix model, representing the state-of-the-art upper bound in accuracy. (ii) **AlphaFold-Multimer (AFM)** (Evans et al., 2021): The v2.3 model, a strong baseline for protein-only complexes. (iii) **CombFold**: A representative "divide-and-conquer" method that predicts all pairwise interactions and assembles them. To ensure a fair comparison, we created a variant, (iv) **CombFold+AF3**: A CombFold variant using our AlphaFold3 as the prediction engine instead of its default AlphaFold-Multimer. (v) **MoLPC**: Another assembly-based method, primarily designed for homomeric complexes. (vi) RoseTTAFold All-Atom: An end-to-end biomolecular complex structure predictor (Krishna et al., 2024). (vii) **Merizo+HIERAFold**: A HierAFold variant in which subunit segmentation was replaced by Merizo, a deep learning based method for domain segmentation on protein chain (Lau et al., 2023), while keeping all other components unchanged. For all methods, the MSA (Edgar & Batzoglou, 2006), baseline details can be found in Appendix A.1.

**Evaluation Metrics** For protein-protein interfaces, we report the **DockQ success rate**, defined as the percentage of predictions with a DockQ score $> 0.23$, following the AlphaFold3 paper (Basu & Wallner, 2016), (Mirabello & Wallner, 2024). For protein-ligand interfaces, we report the **success rate** of predictions achieving a ligand RMSD $\leq 2.0$ Å after alignment to the native pocket, following established benchmarks. We report both the "Oracle" score (the best result among all generated samples) and the "Top-1" score (the result of the single highest-ranked sample).

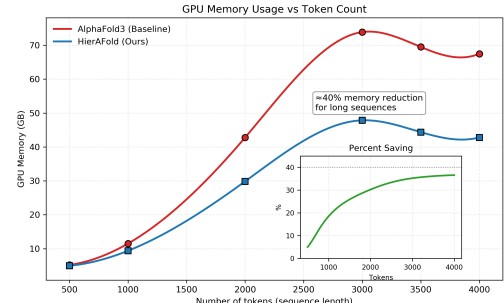

Figure 3: Token number vs GPU memory cost. By default, Protenix inference uses BF16 mixed precision. The decrease when the token number $> 3000$ occurs because SampleDiffusion and ConfidenceHead switch from FP32 to BF16 (set by Protenix). HIERAFOLD saves about 40% GPU memory for large token counts.

## 6.2 RECENT PDB

**Dataset** We used the **recent PDB** benchmark as defined in the Protenix and AlphaFold3 papers. This dataset contains low-homology protein complexes filtered from the Protein Data Bank (PDB) (Bank, 1971) released after the training set cutoff (2021-09-30), providing a robust test of generalization for protein-protein interface prediction, con-

sisting of complexes with up to 20 chains and 5,120 tokens. Following the Alphafold3 and Protenix, for each case, 25 samples were generated using five model seeds (each with five diffusion samples). We report results from our model with 4 recycles and from AFM with 5 recycles.

**Results** As shown in Table 1, HIERAFOLD achieves an Oracle success rate of 73.1% and a Top-1 rate of 69.0% on the recent PDB benchmark. This performance is nearly identical to the end-to-end AlphaFold3 baseline (74.4% Oracle, 70.4% Top-1), demonstrating that our hierarchical approach almost does not sacrifice accuracy on standard-sized complexes. In contrast, both CombFold variants show a significant drop in performance. CombFold+AF3, despite using the same powerful prediction engine, scores only 47.5% (Top-1), highlighting the importance of our method's ability to incorporate multi-chain context during refinement, which purely pairwise methods lack.

The visualizations in Fig. 4 further demonstrate that HIERAFOLD enables more accurate complex prediction. Furthermore, the marginal lower performance of Merizo + HIERAFOLD demonstrates that more complex domain segmentation does not yield better downstream structure prediction and confirms that PAE-based segmentation is well aligned with HierAFold's refinement pipeline without additional costs. More results on subunit segmentation are detailed in Appendix A.9. We further evaluate the peak GPU memory usage compared with the baseline as shown in Fig. 3. Compared with the end-to-end baseline prediction, our method saves $\sim 40\%$ GPU memory when the number of tokens is large, as each inference step only processes a single focus chain and its selected interface subunits.

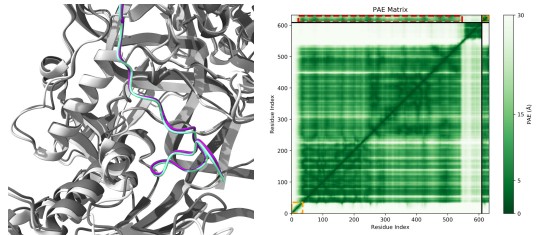

**Performance on Intrinsically Disordered Regions** Intrinsically disordered protein regions (IDRs), sequences lacking stable tertiary structure, are pivotal in cellular processes (Babu et al., 2011; Oldfield et al., 2019). Interactions between IDRs and their partners are characterized by partial and dynamic binding (Tompa & Fuxreiter, 2008) and are accurately captured by AFM (Omidi et al., 2024). To evaluate how IDRs affect HIERAFOLD, we analyzed the recent PDB dataset following established procedures. Each residue was annotated with a disorder probability using AIUPred (Erdős & Dosztányi, 2024), interface residues were identified where any two heavy atoms are within 10 Å of each other (Mirabello & Wallner, 2024) and for each complex we computed the mean disorder score at the interface. Complexes were

Figure 5: Visualization of IDRs prediction. Left: The cyan chain is the IDRs predicted by HIER-AFOLD and the violet chain is the ground truth (PDB ID: 7zwa). Right: The corresponding cropped PAE matrix. The orange square marks divided IDRs subunit (lower left and upper right). The red rectangle highlights the low-PAE region at the IDRs interface.

then grouped into three bins: low (0.1– 0.25), medium (0.25 – 0.5), and high disorder (> 0.5). For both the AlphaFold3 baseline and HIERAFOLD, we measured the mean DockQ for each bin:

| Mean Interface IDR Score | AlphaFold3 (Baseline) Mean DockQ | HIERAFOLD (Ours) Mean DockQ |
|---|---|---|
| Low $(0.1 - 0.25)$ | 0.55 | 0.52 |
| Medium $(0.25 - 0.5)$ | 0.49 | 0.44 |
| High $(> 0.5)$ | 0.46 | 0.41 |

Table 2: Performance comparison on the **recent PDB** dataset, binned by the mean disorder score of interface residues.

The results (Table 2) reveal a consistent trend: as interface disorder increases, DockQ scores decrease for both models. This is expected, because IDRs are flexible, underrepresented in training data, and often participate in transient or fuzzy interactions.

Importantly, HIERAFOLD exhibits almost the **same degradation pattern as AlphaFold3**, demonstrating that our hierarchical coarse-to-fine pipeline preserves AlphaFold3's robustness when handling disordered interfaces. In other words, the introduction of subunit segmentation, interface selection, and hierarchical assembly does not introduce additional weaknesses for IDRs, as shown in

Fig. 5. This robustness stems from how IDRs behave under our PAE-guided workflow: (i) *Segmentation:* IDRs appear as high-PAE or highly flexible regions and are naturally isolated into their own subunits (orange blocks in Fig. 5). (ii) *Interaction Selection:* When IDRs genuinely participate in binding, they produce localized low cross-chain PAE signals that ensure their inclusion during fine-stage refinement (red block in Fig. 5 representing the interaction between KU and PAXX protein). (iii) *Assembly:* Because IDRs generally receive low confidence scores, they are automatically down-weighted during confidence-weighted alignment, preventing unreliable geometry from distorting the global structure.

### 6.3 POSEBUSTER V2

**Dataset** We evaluated protein-ligand performance on the **PoseBuster v2** benchmark (Buttenschoen et al., 2024), a standard dataset for evaluating small molecule docking accuracy containing up to 4,100 tokens and 4 chains.

**Results** On PoseBuster v2, HIERAFOLD achieves a Top-1 success rate of 74.7%, again closely mirroring the 77.4% achieved by the full AlphaFold3 baseline (Table 1). This confirms that our subunit selection and assembly process correctly preserves the fine-grained atomic interactions necessary for accurate ligand placement, and our special handling of small molecular ligands is effective.

### 6.4 FILTERED LARGE COMPLEX DATASET

**Dataset** To test the primary motivation for our work, we curated a dataset of protein complexes filtered from PDB, which was released after 2021-09-30 and contains 5,000 to 10,000 tokens with up to 30 chains. The filtered large complex dataset consists of 664 interfaces. These complexes are too large to be processed on one 80GB A100 GPU by the baseline (AlphaFold3, AlphaFold-Multimer).

**Results** The end-to-end baseline failed with Out-Of-Memory (OOM) errors. HIERAFOLD achieved a Top-1 DockQ success rate of 43.9%, significantly outperforming CombFold+AF3 (19.8%). This result, combined with the visualization in Fig. 6, demonstrates that by intelligently pruning the context for refinement, HIERAFOLD effectively breaks the memory barrier while retaining context needed to resolve complex multi-chain interfaces. The overall performance is lower than on the recent PDB dataset, which can be attributed to two factors. First, the Large Complex Dataset has a much higher proportion of protein-antibody interactions (40% vs. 6%), for which AlphaFold3's performance is known to be lower. Second, the maximum sequence crop size during AlphaFold3 training (768 tokens) is much smaller than most of the chain lengths in this dataset, creating a train-test gap. Despite these challenges, HIERAFOLD still outperforms CombFold by a large margin. Finetuning on a large complex dataset with a larger crop size could potentially mitigate this performance gap.

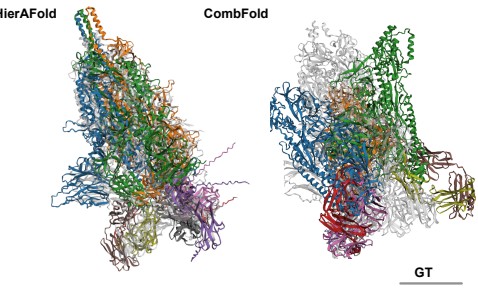

Figure 6: Visualization of our method compared with the CombFold result on the large complex dataset. The PDB ID is 9bj4. The gray chains are ground truth.

## 7 CONCLUSION

We presented HIERAFOLD, a hierarchical framework for predicting large protein complexes that overcomes the memory bottlenecks of existing end-to-end models. By leveraging coarse predictions and PAE-guided modular decomposition, our method automatically identifies subunits and interfaces, and refines each chain in the context of its key interacting partners. This design reduces memory requirements by up to 40% while preserving essential multi-body cooperativity. Experiments show that HIERAFOLD matches AlphaFold3 on standard benchmarks and, critically, extends tractable prediction to complexes exceeding 5,000 tokens, achieving substantial gains over prior divide-and-conquer approaches. Limitations and future work of HIERAFOLD can be found in Appendix A.11.

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

# A  APPENDIX

## A.1  BASELINE DETAILS

In this section, we describe the baseline models employed in our experiments. As most baselines are designed to predict only protein-protein complexes, the evaluation is primarily conducted on the recent PDB dataset.

### A.1.1  MOLPC

We utilized the code from the MoLPC repository. The default settings were applied without modifying any hyperparameters. MoLPC generates a single best predicted protein complex, which was used for evaluation.

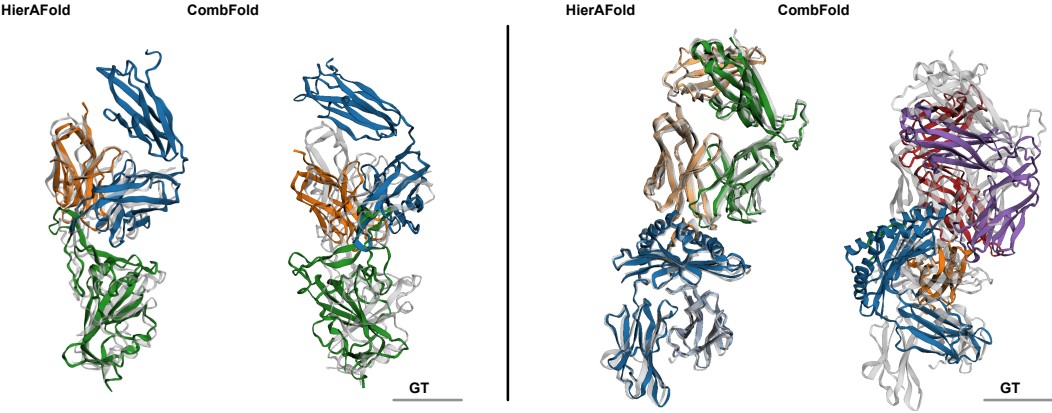

Figure 7: Visualization of complex modeling results comparing HierAFold and CombFold. Left PDB ID is 7wck, and right PDB ID is 7ow5. The gray chains are ground truth.

### A.1.2    COMBFOLD

We employed the code from CombFold repository. CombFold utilizes a combinatorial assembly algorithm to construct protein complexes from their subunits. Its performance relies on the quality and quantity of predicted subset-complexes provided by the underlying protein complex prediction method (AlphaFold-Multimer in default). In this phase, we use both AlphaFold-Multimer v2 from AlphaPulldown repository and Protenix (a reproduction of AlphaFold3) from Protenix repository as the backbone prediction model, and we notate them as CombFold-AFMv2 and CombFold-Protenix. CombFold accepts inputs comprising 2 or more subunits and applies its combinatorial method accordingly, and its performance depends on the number of input. To balance performance and time, we included all heteromeric subunit pairs for every unique combination of distinct subunits, along with homomeric subunit pairs for each chain homomer in the protein complex. Trimeric or longer subunit complexes were not utilized; instead, all possible dimeric subunit complexes served as the starting point for the assembly algorithm.

In CombFold, due to potential inaccuracies in pairwise transformations, subcomplexes are evaluated for steric clashes among backbone atoms with pLDDT scores greater than 80. A subcomplex is filtered out if more than 5% of its atoms exhibit clashes. However, for complexes that fail to yield any valid predictions under this criterion, we disabled the filtering mechanism to generate results. CombFold produces multiple predictions, each accompanied by a confidence score, and we selected the prediction with the highest confidence for our experiments as Top-1 accuracy, and the highest performance within all samples is the oracle accuracy. For the CombFold-AFMv2 baseline, we utilized multiple sequence alignments MSAs from the offline AlphaFold Database. For the CombFold-Protenix baseline, we employed online MSA searches via the Protenix server while performing the complex predictions offline.

### A.2    ALPHAFOLD MULTIMER

We utilized the code from the AlphaPulldown repository. We employed AlphaPulldown's default settings, encompassing multiple sequence alignment (MSA) search and complex prediction. The complex prediction process involves a 5-cycle inference that generates a single prediction, which was utilized in our experiments. The rank-0 prediction sample represents the one with the highest confidence and serves as the Top-1 accuracy metric, while the highest performance among all samples corresponds to the oracle accuracy.

### A.3 ROSETTAFOLD ALL-ATOM

We utilized the code from the RoseTTAFold All-Atom. We employed RFAA's default settings while using the Alphafold-Multimer's Database provided in AlphaPulldown repository v2.0.5. We used Uniref30 version at 2023-02 to replace the original 2020-06 requirement in RFAA.

### A.4 PSEUDO CODE

---

**Algorithm 1** HIERAFOLD: Hierarchical Protein Complex Structure Prediction

---

**Require:** Set of peptide chains $\{C_1, C_2, \ldots, C_M\}$ with total $N$ tokens
**Ensure:** Assembled 3D structure of the protein complex

    **Coarse Stage: Pairwise Prediction and Modular Decomposition**
1: **for** each pair of chains $(C_i, C_j)$, $i < j$ **do**
2:     Generate coarse predictions using distilled model: $\mathbf{X}_{ij}$, $\mathbf{P}_{ij}$ (coordinates and PAE matrix)
3: **for** each chain $C_i$ **do**
4:     Segment $C_i$ into subunits $\{U_{i,1}, U_{i,2}, \ldots\}$ using PAE-guided recursive partitioning on $\mathbf{P}_{ii}$
5: **for** each focus chain $C_a$ **do**
6:     Initialize neighborhood $\mathcal{N}(C_a) = \emptyset$
7:     **for** each other chain $C_b$, $b \neq a$ **do**
8:         **for** each subunit $U_{b,j}$ in $C_b$ **do**
9:             Compute mean interface PAE: $\bar{P}(C_a, U_{b,j})$
10:             Compute min distance: $d_{\min}(C_a, U_{b,j})$
11:             **if** $\bar{P}(C_a, U_{b,j}) < \tau_p$ or $d_{\min}(C_a, U_{b,j}) < \tau_d$ **then**
12:                 Add $U_{b,j}$ to $\mathcal{N}(C_a)$
    **Fine Stage: Interface-Aware Refinement**
13: **for** each focus chain $C_a$ **do**
14:     Form composite input: $C_a \cup \mathcal{N}(C_a)$
15:     Perform full AlphaFold3-style inference to get refined structure $\hat{\mathbf{X}}_a$
    **Assembly Stage: Confidence-Aware Global Alignment**
16: Initialize global assembly with highest-confidence $\hat{\mathbf{X}}_{\text{best}}$
17: **for** each remaining partial structure $\hat{\mathbf{X}}_a$ **do**
18:     Identify overlapping atoms between $\hat{\mathbf{X}}_a$ and global assembly
19:     Compute confidence weights $w_k = \text{pLDDT}(\mathbf{x}_{a,k}) \cdot \text{pLDDT}(\mathbf{x}_{\text{global},k})$
20:     Find optimal $\mathbf{R}$, $\mathbf{t}$ minimizing weighted RMSD:
21: Rank final structures using composite score (ipTM + pTM + clash detection)

---

### A.5 ADDITIONAL COMPARISON WITH COMBFOLD

We further compared HIERAFOLD with CombFold on benchmark 2 provided by CombFold, 25 complexes with 5–30 chains and 2,000-18,000 amino acids. Most of those are included in the recent PDB and filtered large complex data. The top-1 DockQ success rate is 30.2% (CombFold) vs 50.7% (ours).

We further performed an additional analysis to examine how the performance gap between HIER-AFOLD and existing baselines changes as the size of the input complex increases. Specifically, we partitioned complexes in the recent PDB benchmark into bins based on their total token count and computed the Top-1 average DockQ difference between. The result is summarized in Table 3.

### A.6 ABLATION STUDY

To validate the effectiveness of the key components in HIERAFOLD, we conducted a series of ablation studies on the recent PDB dataset. We systematically dismantled our framework to quantify the contribution of the consistency-distilled coarse model, the PAE-based subunit segmentation, the PAE-guided subunit selection, and the confidence-aware assembly mechanism. The results are summarized in Table 4.

| Token Count | HIERAFOLD - CombFold (%) | HIERAFOLD - AlphaFold (%) |
|---|---|---|
| $0 - 1,000$ | +13.4 | -0.2 |
| $1,000 - 2,000$ | +19.4 | -1.0 |
| $2,000 - 3,000$ | +23.0 | -1.4 |
| $3,000 - 4,000$ | +23.9 | -1.4 |
| $> 4,000$ | +23.1 | -1.7 |

Table 3: Analysis of the DockQ success rate gap between HIERAFOLD and CombFold, as well as between HIERAFOLD and AlphaFold, across increasing complex size on the **recent PDB** set. The positive values indicate HIERAFOLD's advantage.

Table 4: Ablation study of HIERAFOLD on the recent PDB dataset. We report the DockQ Success Rate (%) and average inference time for a complex with an average of 3,000 tokens. The results validate that each component of our proposed method contributes to the final performance and efficiency.

| Method | Oracle (%) | Top-1 (%) | Avg. Time (min) |
|---|---|---|---|
| **HIERAFOLD (Ours)** | **73.1** | **69.0** | **46** |
| *Ablation on Coarse Prediction* | | | |
| w/ Full Diffusion Model | 73.3 | 69.4 | 125 |
| w/ Mini-rollout (20 steps) | 71.0 | 68.2 | 52 |
| *Ablation on Subunit Segmentation* | | | |
| $\tau_{split} = 0$ (residue level) | 72.0 | 67.8 | 45 |
| $\tau_{split} = 5$ | 72.9 | 68.9 | 46 |
| $\tau_{split} = 15$ | 73.1 | 69.1 | 46 |
| $\tau_{split} = 20$ | 73.2 | 69.1 | 47 |
| $\tau_{split} = 8$ (on PDE) | 72.4 | 68.2 | 45 |
| *Ablation on Subunit Selection* | | | |
| PAE-Only Selection | 71.0 | 66.9 | 46 |
| Distance-Only Selection | 70.4 | 66.3 | 46 |
| $\tau_p = 10$ and $\tau_d = 30$ | 73.4 | 69.3 | 50 |
| *Ablation on Fine Stage* | | | |
| w Distilled Model | 69.3 | 64.9 | 15 |
| *Ablation on Assembly* | | | |
| w/ Unweighted Assembly | 71.0 | 66.5 | 45 |

**Effect of Distilled Coarse Prediction** The coarse stage is designed to be fast yet accurate enough to guide the decomposition. We tested its importance by replacing it with the full, non-distilled AlphaFold3 model for pairwise predictions. This provides a more accurate but significantly slower coarse structure prediction process. As shown in Table 4, using the full diffusion model offers a marginal 0.2% improvement in Oracle DockQ success rate on the recent PDB dataset but triples the inference time, confirming that our distilled model provides the best trade-off between speed and quality. We further evaluated the mini-rollout method to generate the coarse structure, which degrades the DockQ success rate by 2%, while increasing the inference time, indicating the incorrect pairwise conformation prediction affects the final result.

**PAE-based Segmentation Sensitivity** In HIERAFOLD, a chain is partitioned into subunits based on low-PAE blocks, controlled by the threshold $\tau_{split}$. The default value in our experiments is $\tau_{split} = 10$. Larger thresholds create coarser subunits (fewer boundaries), while smaller thresholds produce finer-grained segmentation. To assess sensitivity to $\tau_{split}$, we systematically varied $\tau_{split}$ and evaluated the downstream complex prediction accuracy (Table 4). The results demonstrate that: (i) Performance remains stable across a broad and practical range of thresholds. (ii) Extremely fine segmentation ($\tau_{split} = 0$) harms performance, confirming that splitting residues individually introduces unnatural discontinuities and disrupts coherent structural modeling. (iii) Very large

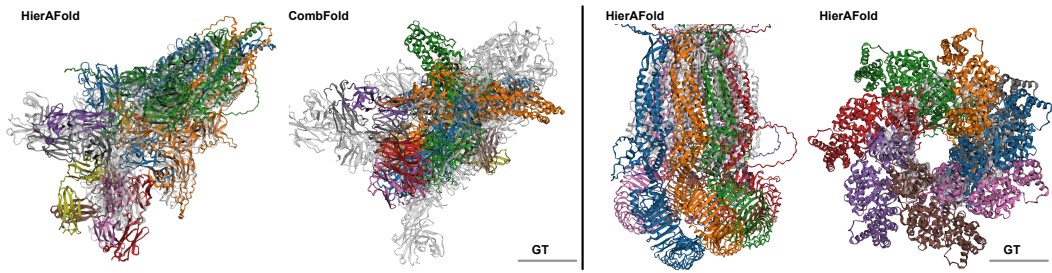

Figure 8: Left: Visualization of large complex modeling results comparing HierAFold and Comb-Fold. PDB ID is 9kt3. Right: Additional visualization of HIERAFOLD on the large protein complex dataset. PDB IDs are 8dxq and 9l2e. The gray chains are ground truth.

thresholds converge to end-to-end behavior, producing single-chain subunits. While accuracy remains stable, memory consumption increases, defeating the scalability benefits of our method.

We further compared PAE with an alternative confidence signal, Predicted Distance Error (PDE), for subunit segmentation. For PDE, we optimized the segmentation threshold to ensure a fair comparison. However, results indicate that PAE better captures pairwise structural coherence essential for identifying rigid subunits, making it the preferred confidence metric for our decomposition procedure.

**Importance of PAE-Guided Subunit Selection** Our core hypothesis is that PAE-guided subunit selection is superior to simpler heuristics. We evaluated this by ablating the selection criteria: (i) PAE-Only: Selecting subunits based only on the mean interface PAE ($\bar{P} < \tau_p$). (ii) Distance-Only: Selecting subunits based only on spatial proximity in the coarse structure ($d_{\min} < \tau_d$). The results show that combining PAE and distance criteria is optimal. The PAE-only approach performs better than the distance-only one, indicating that interaction confidence is a more reliable signal than proximity alone. We further evaluate the effect of a looser selection policy where we changed the $\tau_p$ to 10 and $\tau_d$ to 30 to include more subunits from the other chains. The performance increases marginally with the increased peak memory usage and inference time.

**Contribution of Confidence-Aware Assembly** Finally, we assessed the impact of our confidence-aware alignment algorithm. We compared our standard method against a variant using an Unweighted Assembly, where the Kabsch algorithm is applied without pLDDT weighting. The unweighted method shows a noticeable drop in accuracy (69.0% to 66.5% Top-1). This highlights that weighting the alignment by local confidence makes the assembly process more robust, especially for complexes with flexible linkers or poorly predicted regions, as it correctly prioritizes the well-structured interfaces to guide the overall superposition.

### A.7 MORE VISUALIZATION RESULTS

Fig. 7 shows more visualization of HIERAFOLD on the recent PDB dataset compared with Comb-Fold. Fig. 8 shows HIERAFOLD performance on a large complex dataset, showing improvement in capturing multiple chain interactions and performing well in monomers.

Table 5: DockQ success rate comparison of AlphaFold3 and HIERAFOLD on homomers and heteromers.

| Category | Metric | AlphaFold3 | HIERAFOLD |
|---|---|---|---|
| Homomers | Oracle | 77.3 | 76.2 |
| | Top-1 | 74.0 | 72.8 |
| Heteromers | Oracle | 72.3 | 70.9 |
| | Top-1 | 67.9 | 66.3 |

## A.8    PERFORMANCE ON HOMOMERIC VS. HETEROMERIC COMPLEXES

We further examined performance across homomeric versus heteromeric complexes. Complexes in the **recent PDB** dataset were categorized based on chain composition. As shown in Table 5, both AlphaFold3 and HierAFold achieve higher accuracy on homomers, likely due to stronger co-evolutionary and symmetry-related signals.

subsectionInfluence of the Number of Conformations Our experiments in **recent PDB** follow the standard inference protocol of Protenix (Team et al., 2025), generating 25 conformations. To assess the impact of sampling, we evaluated performance when generating fewer conformations. The results are summarized in Table 6

Table 6: DockQ success rate comparison of AlphaFold3 and HIERAFOLD with different numbers of generated conformations.

| Category | Metric | AlphaFold3 | HIERAFOLD (OURS) |
|---|---|---|---|
| n=5 | Oracle | 69.9 | 68.5 |
| | Top-1 | 65.0 | 63.6 |
| n=10 | Oracle | 72.2 | 70.9 |
| | Top-1 | 67.5 | 66.1 |
| n=25 | Oracle | 74.4 | 73.1 |
| | Top-1 | 70.4 | 69.0 |

## A.9    SUBUNIT SEGMENTATION

To contextualize our approach, we compared the segmentation accuracy between our PAE-based segmentation against Merizo (Lau et al., 2023) on the CATH-663 test set (Sillitoe et al., 2021). As expected, Merizo achieves higher IoU (0.85 vs. 0.65), since it is explicitly designed and trained for domain parsing, including the detection of discontinuous domains that are prevalent in CATH. In contrast, our segmentation module is not optimized for canonical domain detection, yet still achieves reasonable segmentation quality. It deliberately merges tightly packed regions with uniformly low inter-subunit PAE into a single rigid unit—a behavior aligned with our design objective of identifying structurally coherent subunits that matter for downstream refinement, rather than reproducing evolutionary domain boundaries.

Despite this difference in segmentation granularity, our method achieves a boundary MCC of **0.79**, closely matching Merizo's 0.84. This demonstrates that PAE provides a reliable structural signal for locating true physical boundaries, even without specializing in fine-grained domain delineation.

Crucially, these variations in IoU have no negative impact on the final complex prediction accuracy as shown in Table 1. The results confirm that PAE is an effective and sufficient metric for the type of segmentation required by HIERAFOLD—identifying rigid subunits and interaction-relevant boundaries that guide accurate multi-chain refinement.

Combined with the results above, our PAE-based segmentation offers three key advantages:

> **Computational Efficiency:** No additional model or database queries are required. Segmentation reuses PAE matrices already produced during coarse prediction, making it effectively cost-free. **Structural Relevance for Complex Assembly:** PAE identifies rigid, co-operatively moving structural units, not evolutionary domains. It also highlights inter-chain interaction subunits, enabling interface-aware refinement, capabilities absent in existing domain predictors. **Competitive or Better Downstream Performance:** Despite simpler computation, PAE-based segmentation performs better than Merizo when evaluated in the context of complex structure prediction, the task for which HierAFold is optimized.

## A.10    EXTENSION OF HIERAFOLD TO ROSETTAFOLD ALL-ATOM

To evaluate the generalizability of HIERAFOLD beyond AlphaFold3, we integrated RoseTTAFold All-Atom (Krishna et al., 2024) as an alternative backbone predictor. On a representative subset

of 50 complexes from the recent PDB dataset, the the HIERAFOLD-RoseTTAFold configuration achieved a DockQ success rate of 53.7%, closely matching the 54.8% obtained by the end-to-end RoseTTAFold All-Atom model, while reducing memory usage by ∼35%. This demonstrates that HIERAFOLD is model-agnostic: it can seamlessly wrap around different backbone predictors and preserve their accuracy while substantially improving scalability. As stronger predictors emerge, they can be incorporated into the HierAFold pipeline to further enhance performance on large complexes.

### A.11 LIMITATION OF HIERAFOLD

#### A.11.1 COMPUTATION COST AND MITIGATION STRATEGIES

Although HIERAFOLD reduces peak GPU memory usage by decomposing a large complex into manageable subunits, this decomposition introduces extra computation time. We use several strategies to keep this overhead practical: (i) *Distilled Coarse Prediction:* Running full AlphaFold3 for every pairwise coarse prediction would be prohibitively expensive. To address this, we introduce a *consistency-distilled diffusion model* with a shallower Pairformer. This replacement significantly reduces latency in the coarse stage while maintaining effective structural priors, as shown in Table 4. (ii) *Parallelized Refinement:* For a complex with $M$ chains, HIERAFOLD runs the fine-stage refinement $M$ times–once for each focus chain–but each run operates on a substantially smaller input, lowering per-run memory. This reduction allows: parallel execution across multiple GPUs, or *batching execution* in the diffusion module on a single GPU. These options keep total inference time manageable while enabling prediction of complexes far beyond AlphaFold3's memory limit. We benchmarked both methods on a single A100 GPU. The results are shown in Table 7.

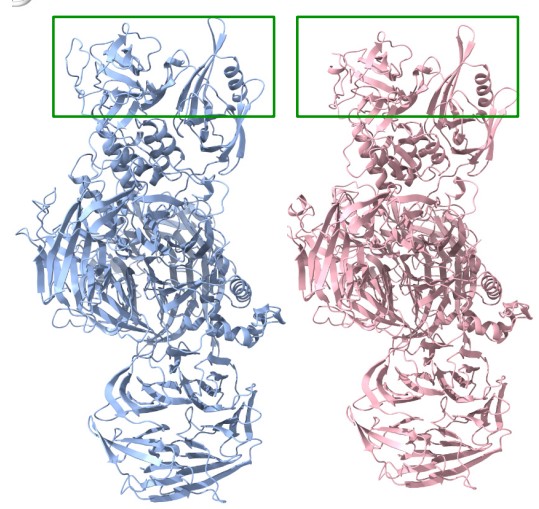

Figure 9: Conformation coverage is limited. The left shows the prediction for PDB 8cvp (apo), and the right for 8d7u (holo). The apo structure should adopt an open conformation, while the holo is closed. Both are predicted in the closed state in HIERAFOLD (green box)

| Total Tokens | AlphaFold3 (min) | HIERAFOLD (min) |
|---|---|---|
| 2,000 | 14 | 17 |
| 3,000 | 38 | 46 |
| 4,000 | 74 | 98 |
| 5,000 | OOM | 170 |

Table 7: Inference time comparison. While HIERAFOLD introduces extra computation cost, it successfully runs on large complexes where the baseline fails due to memory constraints. The baseline inference time is measured on Protenix v0.5.0.

#### A.11.2 MULTI-STATE ASSEMBLIES

A known challenge for AlphaFold-family models is the prediction of distinct functional states. Since HIERAFOLD uses AlphaFold3 as its predictor, it inherits this limitation. For example, when predicting the structures of an E3 ubiquitin ligase in its apo (PDB: 8cvp, expected open state) and holo (PDB: 8d7u, expected closed state) forms, HIERAFOLD, like the baseline, exclusively predicts the closed conformation for both (Fig. 9). Methods developed for AlphaFold-Family that assist in

multistate prediction may be adaptable in HIERAFOLD (Wayment-Steele et al., 2024; Heo & Feig, 2022).

### A.11.3 PERFORMANCE BOUND AND FUTURE WORK

Despite these advances, performance remains bounded by the underlying AlphaFold3 model, particularly for very large or challenging complexes. Future work will explore training strategies and model enhancements to further improve accuracy on such complex tasks.

### A.12 LLM USAGE

Throughout the preparation of this manuscript, a Large Language Model (LLM) was utilized as a writing assistant to enhance the quality of the text. The primary role of the LLM was to polish the language by improving grammar, refining sentence structure for clarity, and ensuring stylistic consistency. It is important to state that the LLM's contribution was strictly limited to language enhancement. All conceptual development, research, analysis, and the core arguments presented in this paper are the original work of the authors. Following the guidelines on academic integrity, we have critically reviewed and edited all machine-generated suggestions and take full responsibility for the final content and scientific accuracy of this publication.

