# OpenReview forum: "Efficient Prediction of Large Protein Complexes via Subunit-Guided Hierarchical Refinement"
_ICLR.cc/2026/Conference — ICLR 2026 Poster_

### Official Review · Reviewer_XQxJ · 2025-10-23

**Soundness:** 3
**Presentation:** 2
**Contribution:** 3
**Rating:** 4
**Confidence:** 4

**Summary:**

This paper addresses the computational complexity issue in AlphaFold by presenting a hierarchical pipeline, refered to as HieraFold, which decomposes the end-to-end structure prediction task in a coarse-to-fine manner.
HieraFold first performs a coarse global prediction using a "lightweight" version of AlphaFold with a smaller diffusion module, and then locally refines critical subunits identified via the pAE matrix.
Experimental results on protein-protein and protein-ligand benchmarks demonstrate the effectiveness of the proposed method.

**Strengths:**

1. This paper addresses an important research problem: reducing the peak computational complexity of AlphaFold, which is critical for real-world applications involving very large protein complexes.

2. The proposed method for identifying critical subunits does not rely on expert curation, making it more generalizable and practical.

3. Experimental results on large complexes demonstrate substantial improvements over existing methods.

**Weaknesses:**

1. **Writing and organization.** Although generally well structured, the manuscript would benefit from improvements in writing. Specific issues include:
   * Incorrect use of hyphens and dashes;
   * Some paragraphs are overly wordy;
   * The Introduction section discusses the literature extensively but there is no separate “Related Works” section. It is important to maintain the distinct roles of these sections rather than combining them;
   * Line 249: “focus chai” should be corrected to “focus chain.”

2. **Computational and time cost.** The multi-stage pipeline leads to higher computational overhead and longer runtime. A more detailed analysis of the trade-off between the ability to handle large complexes and the additional time and computation required would strengthen the paper.

3. **Insufficient validation of design choices.** Some key design decisions are not thoroughly supported by ablation studies, including:
   * The choice of pAE over other confidence metrics output by AlphaFold;
   * Whether confidence-metric-based subunit identification indeed outperforms manually designed approaches;
   * The necessity of the refining stages (Stages 2 and 3), i.e., how much improvement does the full pipeline provide over the coarse stage alone.

**Questions:**

1. A fundamental question: why do low-pAE regions naturally correspond to critical subunits? Please provide an intuitive explanation of the rationale behind this choice.

2. In Figure 2, why is C1 retained while C2 and C3 are omitted?

3. How does the performance gap between HieraFold and the baseline change as the size of the input complex increases?

---

> ### Author Response · Authors · 2025-11-23
>
> Dear Reviewer XQxJ,
>
> Thank you for your constructive feedback and recognition. We have added some experiments and revised the manuscript according to your comments.
>
> ### **Weakness 1: Writing and organization**
>
> Thank you for these helpful suggestions. We have revised the manuscript accordingly.
>
> - **Incorrect use of hyphens and dashes:** We carefully reviewed all hyphens, en dashes, and em dashes throughout the manuscript and corrected inconsistent or improper usage.
> - **Overly wordy paragraphs:** Several sections—particularly those introducing AlphaFold3 and its confidence representations—have been rewritten for clarity and conciseness.
> - **Missing Related Works section:** We added a dedicated **Section 2: Related Works**, and relocated the detailed literature discussion from the Introduction to this new section to clearly separate background motivation from prior work.
> - **Spelling correction:** The typo “focus chai” has been corrected to “focus chain” (line 249). Thank you for catching this.
>
> ### **Weakness 2: Computational and time cost**
>
> Although **HierAFold** substantially reduces peak GPU memory usage by partitioning large complexes into manageable subunits, this decomposition introduces extra computational overhead. We implement several strategies to mitigate this cost:
>
> 1. **Distilled Coarse Prediction**
>
>     A naive coarse stage that invokes the full AlphaFold3 model for every pairwise prediction would be prohibitively slow. To avoid this, we train a **consistency-distilled diffusion model** with a shallower Pairformer. This significantly accelerates the coarse stage while providing strong structural priors. Ablations in Table 4 (revised manuscript) demonstrate substantial latency reduction.
>
> 2. **Parallelized or Batched Fine-Stage Refinement**
>
>     For a complex with M chains, **HierAFold** performs the refinement stage M times—once for each focus chain. However, each refinement pass operates on a **much smaller input**, reducing per-run memory usage. This enables:
>
>     - **parallel execution** on multi-GPU systems, or
>     - **batched execution** in the diffusion module on a single GPU.
>
>     As a result, total inference time remains comparable of that of AlphaFold3, while enabling successful prediction of complexes far larger than those supported by AlphaFold3.
>
>
> We provide reference inference times on a single A100 GPU in Table 1:
>
> | **Tokens** | **AF3 (min)** | **HierAFold (min)** | **AF3 (GB)** | **HierAFold (GB)** |
> | --- | --- | --- | --- | --- |
> | 2,000 | 14 | 17 | 42.8 | 30.1 |
> | 3,000 | 38 | 46 | 74.5 | 48.7 |
> | 4,000 | 74 | 98 | 69.3 | 44.3 |
> | 5,000 | OOM | 170 | OOM | 60.7 |
>
> *Table 1: Inference time comparison. While **HierAFold** introduces extra computation cost, it successfully runs on large complexes where the baseline fails due to memory constraints. The baseline inference time is measured on Protenix v0.5.0.*
> ### **Weakness 3: Insufficient validation of design choices**
>
> Thank you for this valuable suggestion. We have added the requested ablation studies and expanded our analysis in the revised manuscript. Below we summarize the new experiments and the rationale behind each design decision.
>
> **1.  The choice of PAE over other confidence metrics output by AlphaFold**
>
> We conducted an ablation study comparing **Predicted Aligned Error (PAE)** with an alternative confidence signal, **Predicted Distance Error (PDE)**, for subunit segmentation. For PDE, we optimized the segmentation threshold to ensure a fair comparison. As shown in Table 2, **PAE consistently outperforms PDE** for both Oracle and Top-1 DockQ scores.
>
> | **Method** | **Oracle (%)** | **Top-1 (%)** |
> | --- | --- | --- |
> | \tau_{split}=10 (on PAE) | 73.1 | 69.0 |
> | \tau_{split}=8 (on PDE) | 72.4 | 68.2 |
>
> *Table 2: Ablation study about segmentation value and dependency metric.*
>
> These results indicate that **PAE better captures pairwise structural coherence essential for identifying rigid subunits**, making it the preferred confidence metric for our decomposition procedure.
>
> **2. Whether confidence-metric-based subunit identification indeed outperforms manually designed approaches**
>
> We expanded the **Related Work** section to discuss manually designed domain and subunit segmentation methods and conducted empirical comparisons against state-of-the-art approaches such as **Merizo** and domain-based annotations (CATH/ECOD). As detailed in **Common Concern 1**, these experiments show that:
>
> - Although manual or domain-based methods may achieve higher domain-level segmentation IoU,
> - **PAE-based segmentation yields higher downstream complex prediction accuracy**,
> - highlighting that **rigidity-based segmentation aligns better with the needs of hierarchical refinement** than evolutionary or predefined domain boundaries.
>
> This validates that **confidence-based subunit identification is not only effective but also better aligned with our target task**.

---

> ### Author Response · Authors · 2025-11-23
>
> **3. The necessity of the refining stages (Stages 2 and 3)**
> Stages 2 and 3 are critical components of **HierAFold**. The coarse stage uses a **distilled diffusion model**, which trades accuracy for speed. Without refinement, the predictions exhibit noticeable quality degradation. To quantify this, we ran an ablation where the refinement stages were removed:
>
> | **Method** | **Oracle (%)** | **Top-1 (%)** | **Avg. Time (min)** |
> | --- | --- | --- | --- |
> | **HierAFold (Ours)** | **73.1** | **69.0** | **46** |
> | w/ Distilled Model | 69.3 | 64.9 | 15 |
>
> *Table 3: Ablation study about the necessity of the fine stage.*
>
> This evaluation shows that **the fine stages contribute +3–4% absolute DockQ improvement**, recovering errors introduced by the distilled coarse predictor and enabling accurate local and inter-chain refinement. The full pipeline is therefore necessary to achieve competitive performance with AlphaFold3 while retaining scalability.
>
> Across all three design decisions, the new experimental results strongly support the architectural choices in **HierAFold**:
>
> - **PAE** is the most effective confidence metric for segmentation.
> - **Confidence-based segmentation** is better aligned with hierarchical refinement than domain-based alternatives.
> - **Refinement stages** are essential to restore accuracy lost in the coarse stage and achieve high-quality predictions.
>
> We have incorporated these analyses into the revised manuscript.
>
> ### **Question 1: Why do low-PAE regions naturally correspond to critical subunits?**
>
> **Predicted Aligned Error (PAE)** estimates the model’s expected positional uncertainty between residue pairs. Intuitively, when two regions of a protein **move coherently as a rigid unit**, the model can predict their *relative* positions with high confidence, leading to **low PAE**. Conversely, residues belonging to different domains or flexible linkers exhibit **higher relative uncertainty** and therefore **higher PAE**.
>
> Thus, low-PAE blocks naturally highlight **structurally cohesive, rigid subunits**, while high-PAE transitions indicate **domain or subunit boundaries**.
>
> ---
>
> ### **Question 2: In Figure 2, why is C1 retained while C2 and C3 are omitted?**
>
> Thank you for pointing it out. We acknowledge that the original diagram was unclear. In Figure 2, **C1 is retained** because it satisfies our interaction-selection criteria: it has **low cross-chain PAE** and is **spatially close** to the focus chain, indicating a high-confidence, structurally relevant interaction. In contrast, **C2 and C3 are omitted** because they either exhibit **high PAE** with respect to the focus chain (low interaction confidence) or are **too distant** to be considered meaningful interaction partners.
>
> We have revised Figure 2 to make these criteria and the selection process clearer.
>
> ---
>
> ### **Question 3: How does the performance gap between HierAFold and the baseline change as the size of the input complex increases?**
>
> Thank you for the suggestion. We performed an additional analysis to examine how the performance gap between **HierAFold** and existing baselines changes as the size of the input complex increases. Specifically, we partitioned complexes in the recent PDB benchmark into bins based on their total token count and computed the **Top-1 average DockQ difference** between:
>
> - **HierAFold vs. CombFold**, and
> - **HierAFold vs. AlphaFold3**.
>
> The results are summarized in Table 4.
>
> | **Token Count** | **HierAFold - CombFold (%)** | **HierAFold - AlphaFold (%)** |
> | --- | --- | --- |
> | 0–1,000 | +13.4 | -0.2 |
> | 1,000–2,000 | +19.4 | -1.0 |
> | 2,000–3,000 | +23.0 | -1.4 |
> | 3,000–4,000 | +23.9 | -1.4 |
> | >4,000 | +23.1 | -1.7 |
>
> *Table 4: Analysis of the DockQ success rate gap between **HierAFold** and CombFold, as well as between **HierAFold** and AlphaFold, across increasing complex size on the recent PDB set. The positive values indicate **HierAFold**'s advantage.*
>
> The analysis shows that **HierAFold’s advantage over CombFold increases steadily with complex size**, demonstrating the benefit of our scalable hierarchical refinement pipeline. Compared to AlphaFold3, **HierAFold maintains a nearly constant and small performance gap**, while retaining the ability to process complexes far beyond AlphaFold3’s memory limits.
>
> To address the reviewers' concerns, we have conducted several additional experiments and analyses, including:
>
> - Ablation studies validating the superiority of PAE over PDE for segmentation and the necessity of refinement stages.
> - A detailed analysis of computational costs and the performance gap relative to the complex size.
> - Comparisons with other segmentation methods trained on manually labelled domains.
> - Significant improvements to the manuscript's writing and organization, including the addition of a dedicated Related Works section.
>
> All revisions in the paper are marked in blue font. We appreciate the reviewers' constructive suggestions and remain committed to improving our work.

---

> > ### Comment · Reviewer_XQxJ · 2025-11-26
> > **Post Rebutall**
> >
> > Thanks to the authors for addressing my questions. I will raise my rating to 6.

---

> > > ### Author Response · Authors · 2025-11-26
> > >
> > > Dear Reviewer XQxJ,
> > >
> > > We thank the reviewer for acknowledging our work. We're happy to answer any additional questions.

---

### Official Review · Reviewer_E9v3 · 2025-10-28

**Soundness:** 2
**Presentation:** 3
**Contribution:** 2
**Rating:** 4
**Confidence:** 2

**Summary:**

This paper presents HIERAFOLD, which exploits the modularity of large complexes via PAE-guided (Predicted Aligned Error) subunit decomposition, targeted interface-aware refinement, and confidence-weighted assembly. By leveraging coarse predictions and PAE-guided modular decomposition, the method automatically identifies subunits and interfaces, and refines each chain in the context of its key interacting partners. Experiments from various benchmarks clearly show that HIERAFOLD matches AlphaFold3 on standard benchmarks and, critically, extends tractable prediction to complexes exceeding 5,000 tokens, achieving substantial gains over prior divide-and-conquer approaches.

**Strengths:**

1. The challenges and motivation of the proposed method are clearly shown and demonstrated.

2. Various experiments are conducted to show the effectiveness of the proposed method.

3. The paper is well written and organized.

**Weaknesses:**

1. The decomposition is a good way to save memory usage. However, the paper only uses AlphaFold3 as the backbone model to test the method. An extension to other backbone methods will help better demonstrate the effectiveness.

2. More baselines should be added, such as OpenFold and MoLPC, to demonstrate the effectiveness of the proposed methods better.

3. The code is not publicly available.

**Questions:**

1. What is the computation cost of HIERAFOLD? Though the decomposition of long tokens into several small subunits is effective at reducing the memory cost. It ends up with a lot of extra computation. Is the extra computation cost acceptable compared to the saved memory usage?

2. How will the split k in PAE-Guided Subunit Segmentation for Biological Modularity affect the performance? And how much difference for the split part? Is the split showing a significant difference between the two split parts?

---

> ### Author Response · Authors · 2025-11-23
>
> Dear Reviewer E9v3,
>
> We sincerely thank the reviewer for the constructive feedback and for recognizing **HierAFold**. We have revised the manuscript to address your concerns regarding computational cost, parameter sensitivity, and additional baselines. Below is our detailed response based on new experiments.
>
> ### **Weakness 1: An extension to other backbone methods will help better demonstrate the effectiveness.**
>
> Thank you for this valuable suggestion. To evaluate the generalizability of **HierAFold** beyond AlphaFold3, we integrated **RoseTTAFold All-Atom** as an alternative backbone predictor. On a representative subset of 50 complexes from the recent PDB dataset, the HierAFold–RoseTTAFold configuration achieved a **DockQ success rate of 53.7%**, closely matching the **54.8%** obtained by the end-to-end RoseTTAFold All-Atom model, while reducing memory usage by approximately **35%**.
>
> These results demonstrate that **HierAFold is model-agnostic**: it can seamlessly wrap around different backbone predictors and preserve their accuracy while substantially improving scalability. As stronger predictors emerge, they can be incorporated into the HierAFold pipeline to further enhance performance on large complexes.
>
> ### **Weakness 2: More baselines should be added, such as OpenFold and MoLPC.**
>
> Thank you for this helpful suggestion. We have updated the manuscript to clarify our baseline coverage and to further demonstrate the effectiveness of **HierAFold**.
>
> - **MoLPC**: A comparison with MoLPC was *already included* in the original submission. The results are reported in **Table 1** of the manuscript, where **HierAFold** consistently outperforms MoLPC across the evaluated metrics.
> - **OpenFold**: OpenFold is an open-source reimplementation of AlphaFold2, and for complex prediction it directly relies on the **pretrained AlphaFold-Multimer v2.3 parameters**, as stated in its official documentation. Since our manuscript already includes **AlphaFold-Multimer (AFM)** as a baseline (Table 1), the expected performance of OpenFold is essentially identical to AFM. We have clarified this equivalence in the revised text.
> - **Additional Backbone Evaluation**: To further strengthen our empirical evidence, we implemented **HierAFold using RoseTTAFold All-Atom** as the backbone predictor. The results—now included in **Table 1** of the revised manuscript—show a **top-1 DockQ success rate of 55.4%** on the recent PDB dataset, confirming that **HierAFold is effective across different backbone models**.
>
> Together, these results reinforce that **HierAFold consistently improves scalability and maintains high predictive quality across multiple structure predictors**, demonstrating its model-agnostic applicability.
>
> ### **Weakness 3: The code is not publicly available.**
>
> We will make the code, including the training stage and inference stage, public after the paper is published.

---

> ### Author Response · Authors · 2025-11-23
>
> ### **Question 1: Computation cost of HierAFold**
>
> Thank you for raising this important question. We have added a detailed analysis of computational cost and mitigation strategies in **Appendix A.12.1** of the revised manuscript. While **HierAFold** introduces additional computation compared to a single end-to-end AlphaFold3 pass, this overhead is acceptable—and necessary—because **it enables structure prediction for complexes that cause Out-Of-Memory (OOM) failures in the baseline**, a capability fundamentally unavailable to existing predictors. Below is the new paragraph added to the manuscript.
>
> Although **HierAFold** reduces peak GPU memory usage by decomposing a large complex into manageable subunits, this decomposition introduces extra computation time. We use several strategies to keep this overhead practical:
>
> 1. **Distilled Coarse Prediction**
>
>     Running full AlphaFold3 for every pairwise coarse prediction would be prohibitively expensive. To address this, we introduce a **consistency-distilled diffusion model** with a shallower Pairformer. This replacement significantly reduces latency in the coarse stage while maintaining effective structural priors (see Table 4 of the revised manuscript).
>
> 2. **Parallelized Fine-Stage Refinement**
>
>     For a complex with M chains, HierAFold runs the fine-stage refinement M times—once per focus chain—but **each run operates on a substantially smaller input**, lowering per-run memory. This reduction allows:
>
>     - **parallel execution** across multiple GPUs, or
>     - **batched execution** in the diffusion module on a single GPU.
>
>         These options keep total inference time manageable while enabling prediction of complexes far beyond AlphaFold3’s memory limit.
>
>
> We benchmarked both methods on a single A100 GPU. The results are shown below:
>
> | **Tokens** | **AF3 (min)** | **HierAFold (min)** | **AF3 (GB)** | **HierAFold (GB)** |
> | --- | --- | --- | --- | --- |
> | 2,000 | 14 | 17 | 42.8 | 30.1 |
> | 3,000 | 38 | 46 | 74.5 | 48.7 |
> | 4,000 | 74 | 98 | 69.3 | 44.3 |
> | 5,000 | OOM | 170 | OOM | 60.7 |
>
> *Table 1: Inference time comparison. While **HierAFold** introduces extra computation cost, it successfully runs on large complexes where the baseline fails due to memory constraints. The baseline inference time is measured on Protenix v0.5.0.*
>
> ### **Question 2: How will the split k in PAE-Guided Subunit Segmentation for Biological Modularity affect the performance?**
>
> Thank you for the question. We conducted a detailed sensitivity analysis of the PAE-based segmentation split threshold (\tau_{split}) in **Appendix A.6** and the **Table below**. The results show that **HierAFold’s performance remains highly stable across a broad and biologically reasonable range of split thresholds**, indicating that the precise choice of k (i.e., the granularity of the split) does not significantly affect downstream structure prediction. This suggests that the subunits generated by different threshold settings are structurally similar in practice, and the method is robust to variations in the split boundary.
>
> | **PAE Split Threshold (\tau_{split})** | **Oracle DockQ (%)** | **Top-1 DockQ (%)** |
> | --- | --- | --- |
> | 0 (Residue-level selection) | 72.0 | 67.8 |
> | 5 | 72.9 | 68.9 |
> | **10 (Our Default)** | 73.1 | 69.0 |
> | 15 | 73.1 | 69.1 |
> | 20 | **73.2** | **69.1** |
>
> *Table 2: Sensitivity analysis of the PAE split threshold on the recent PDB benchmark. Performance is stable within a reasonable threshold range.*
>
> In detail, the results demonstrate that:
>
> 1. **Performance remains stable across a broad and practical range of thresholds**.
>
>     This indicates that the PAE-guided decomposition does not introduce harmful structural breaks within realistic operating ranges.
>
> 2. **Extremely fine segmentation (**\tau_{split}**=0) harms performance**, confirming that splitting residues individually introduces unnatural discontinuities and disrupts coherent structural modeling.
> 3. **Very large thresholds converge to end-to-end behavior**, producing single-chain subunits. While accuracy remains stable, memory consumption increases, defeating the scalability benefits of our method.
>
> These observations collectively suggest that **PAE naturally reflects coherent structural units**, and segmentation boundaries derived from it do not introduce detrimental artifacts.
>
> To address the reviewers' concerns, we have conducted several additional experiments and analyses, including:
>
> - Validation of the framework's generalizability by integrating RoseTTAFold All-Atom as an alternative backbone predictor.
> - A detailed analysis of computational costs and mitigation strategies.
> - A sensitivity analysis of the PAE split threshold.
> - Clarification regarding additional baselines (OpenFold, MoLPC) and code availability.
>
> All revisions in the paper are marked in blue font. We appreciate the reviewers' constructive suggestions and remain committed to improving our work.

---

> ### Comment · Reviewer_E9v3 · 2025-11-25
>
> Thanks for addressing my concerns. I'd like to maintain my score.

---

> > ### Author Response · Authors · 2025-11-25
> >
> > Dear reviewer E9v3,
> >
> > Thanks again for your time in reviewing our paper. I want to kindly ask if you have any additional questions or concerns. I’m happy to provide any further explanations or revisions.

---

### Official Review · Reviewer_EYwv · 2025-10-30

**Soundness:** 2
**Presentation:** 3
**Contribution:** 1
**Rating:** 4
**Confidence:** 5

**Summary:**

HIERAFOLD: Hierarchical pipeline for efficient large complex prediction using PAE decomposition & refinement.

**Strengths:**

The work addresses a critical and well-known scalability bottleneck in modern protein structure prediction models, which suffer from quadratic complexity (O(N^2)) for large protein complexes. The originality of the method is high, centered on the Predicted Aligned Error (PAE)-guided hierarchical decomposition of large complexes into smaller, manageable subunits. This is a creative, biologically motivated, and highly relevant solution that directly exploits the inherent modularity of protein structures. The quality of the method is evidenced by the thoughtful design, particularly the targeted interface-aware refinement strategy, which focuses computational power precisely where inter-chain interactions are most critical, thus optimizing the accuracy-efficiency trade-off. The significance of HIERAFOLD is substantial, as it practically extends the application of highly accurate prediction methods to systems exceeding a few thousand residues, which is a major advancement for predicting large, biologically relevant assemblies.

**Weaknesses:**

1.Sensitivity to Initial Subunit Quality：The hierarchical pipeline is fundamentally limited by the quality of the initial prediction for the subunits. The current approach implicitly assumes the intra-subunit structure is largely correct and focuses its efforts on inter-subunit interfaces. The method lacks a mechanism to effectively identify or recover from errors in the internal structure prediction of the subunits themselves. This poses a significant failure mode, as local structural errors within a subunit cannot be easily rectified during the refinement or re-assembly steps, especially if they are far from the interface.

2.Decomposition Robustness and Structural Breaks: While efficient, the PAE-based decomposition strategy carries a risk of introducing structural discontinuities or breaks at the boundaries of the artificially created subunits, particularly in cases of highly dynamic, ambiguous, or tightly interwoven (non-modular) interfaces. This potential for introduced structural artifacts needs a more robust discussion and mitigation strategy.

**Questions:**

1.Addressing Intra-Subunit Errors: Given that the method assumes correct intra-subunit structure, how would HIERAFOLD perform if the base model prediction for one of the subunits was highly inaccurate (e.g., due to novel topology or unusual folding)? Could the targeted refinement be adapted to dedicate a small amount of computational budget for intra-subunit refinement based on a local confidence score (e.g., pLDDT)?

2.Decomposition Sensitivity Analysis: How is the threshold for PAE-guided subunit decomposition determined? Is it a fixed value, or is it dynamically tuned? Please provide a sensitivity analysis showing the final accuracy (e.g., TM-score, Interface TM-score) as a function of this PAE decomposition threshold to address the concern about structural breaks.

3.Assembly Mechanism for Flexibility: Can the "confidence-weighted assembly" mechanism explicitly handle the difference between rigid-body movements of subunits (which require global transformation) and local, flexible changes (which require local backbone/side-chain adjustments)? How does the assembly process prevent structural strain introduced during the re-assembly of independently refined subunits?

**Details Of Ethics Concerns:**

no significant ethical concerns

---

> ### Author Response · Authors · 2025-11-23
>
> Dear Reviewer EYwv,
>
> We sincerely appreciate your constructive feedback and acknowledgment of our work. Below, we respond to the weaknesses and questions with new data and analyses incorporated into the revised manuscript.
>
> ### **Weakness 1 and Question 1: Sensitivity to Initial Subunit Quality**
>
> We thank the reviewer for this insightful comment. We agree that the quality of the initial coarse subunits is important and have clarified how **HierAFold** addresses this concern.
>
> **1. Fine Stage explicitly revises and corrects subunit errors.**
>
> While the coarse stage provides an initial decomposition, the **Fine Stage** does *not* assume that intra-subunit geometry is correct. Each focus chain undergoes a **full AlphaFold3 inference**, together with its selected interacting contexts. This stage recomputes all structural features from scratch and is capable of:
>
> - correcting local structural errors inherited from the coarse prediction,
> - recovering misfolded regions, and
> - folding novel topologies when supplied with relevant interaction partners.
>
> Thus, the hierarchical refinement does not simply “freeze” subunit structures but explicitly re-optimizes them.
>
> **2. Failure cases where the base predictor misfolds a subunit are inherent model limitations.**
>
> If the underlying structure predictor produces a fundamentally incorrect fold for a subunit—even when given full AlphaFold3 inference during the Fine Stage—then the resulting errors propagate to assembly. In such cases, the failure is due to the **limitations of the base predictor**, not the hierarchical framework itself. Our work focuses on overcoming the *computational* barriers that prevent end-to-end inference on large complexes; we therefore operate under the standard assumption that the predictor is generally capable of folding individual chains.
>
> Importantly, **HierAFold does not restrict or weaken the representational capacity of the predictor**; it only restructures the inference process to make it scalable.
>
> **3. The framework is fully modular and can integrate more accurate future predictors.**
>
> If a stronger single-chain or multi-chain base model is developed in the future, it can be plugged into the HierAFold pipeline without modification. We validated this by integrating **RoseTTAFold All-Atom** into our framework, achieving a DockQ success rate of **53.7%**, comparable to its end-to-end performance of **54.8%**, while reducing memory consumption by ~35% (Appendix A.11). This demonstrates that the pipeline preserves predictive power while offering computational advantages.
>
> **4. Empirical robustness analysis using IDRs.**
>
> To further evaluate the robustness of HierAFold to structurally ambiguous or error-prone regions—analogous to intra-subunit misfolds—we conducted a dedicated analysis on **intrinsically disordered regions (IDRs)**. Despite their inherent unpredictability, **HierAFold matches AlphaFold3’s performance trends across low-, medium-, and high-disorder interfaces**, showing no additional degradation introduced by our hierarchical approach. Detailed results are provided in **Common Concern 2**.
>
> In summary, while the hierarchical design relies on the base predictor’s ability to fold individual chains, **HierAFold actively refines and corrects subunit structures during the Fine Stage**, maintains compatibility with stronger future predictors, and exhibits robust performance even on structurally flexible or ambiguous regions.

---

> ### Author Response · Authors · 2025-11-23
>
> ### **Weakness 2 and Question 2: Decomposition Robustness and Structural Breaks**
>
> We thank the reviewer for this critical and constructive feedback regarding the robustness of PAE-guided decomposition and the risk of introducing structural discontinuities at subunit boundaries. We have substantially expanded our analysis and include a detailed sensitivity study on the PAE split threshold (\tau_{split}). to quantify and mitigate this concern.
>
> **1. Sensitivity of PAE-guided Decomposition**
>
> In **HierAFold**, a chain is partitioned into subunits based on low-PAE blocks, controlled by the threshold \tau_{split}. The default value in our experiments is \tau_{split}=10. Larger thresholds create coarser subunits (fewer boundaries), while smaller thresholds produce finer-grained segmentation.
>
> To assess whether segmentation boundaries introduce structural breaks, we systematically varied \tau_{split} and evaluated the downstream complex prediction accuracy (Table 1). The results demonstrate that:
>
> 1. **Performance remains stable across a broad and practical range of thresholds**.
>
>     This indicates that the PAE-guided decomposition does not introduce harmful structural breaks within realistic operating ranges.
>
> 2. **Extremely fine segmentation (**\tau_{split}**=0) harms performance**, confirming that splitting residues individually introduces unnatural discontinuities and disrupts coherent structural modeling.
> 3. **Very large thresholds converge to end-to-end behavior**, producing single-chain subunits. While accuracy remains stable, memory consumption increases, defeating the scalability benefits of our method.
>
> These observations collectively suggest that **PAE naturally reflects coherent structural units**, and segmentation boundaries derived from it do not introduce detrimental artifacts, provided that τsplit\tau_{\text{split}}τsplit is chosen within a reasonable interval.
>
> | **PAE Split Threshold (\tau_{split})** | **Oracle DockQ (%)** | **Top-1 DockQ (%)** |
> | --- | --- | --- |
> | 0 (Residue-level selection) | 72.0 | 67.8 |
> | 5 | 72.9 | 68.9 |
> | **10 (Our Default)** | 73.1 | 69.0 |
> | 15 | 73.1 | 69.1 |
> | 20 | **73.2** | **69.1** |
>
> *Table 1: Sensitivity analysis of the PAE split threshold on the recent PDB benchmark. Performance is stable within a reasonable threshold range.*
>
> **2. Additional Mitigation via Interface Selection and Fine-Stage Refinement**
>
> It is also important to highlight two additional mechanisms that prevent structural artifacts from propagating:
>
> - **Interface-aware selection:**
>
>     Even if a boundary is imperfect, the system selects interacting residues using *both* cross-chain PAE and predicted distance, ensuring that interwoven or ambiguous interfaces are included in the refinement stage.
>
> - **Full AlphaFold3 refinement per subunit:**
>
>     Each focus chain undergoes a full AlphaFold3 inference with its selected partners. This enables the model to **repair local discontinuities**, refine backbone geometry around segmentation boundaries, and correct any coarse-level artifacts introduced by segmentation.
>
>
> These two components ensure that segmentation does not lock the system into structurally inconsistent configurations.
>
> **3. Comparison to Other Segmentation Methods**
>
> We also provide a quantitative comparison against established domain/subunit segmentation tools (e.g., Merizo), discussed in **Common Concern 1**.
>
> These experiments show that:
>
> - even if our segmentation is coarser,
> - **downstream structure prediction accuracy is higher**,
> - demonstrating that PAE captures the structural modularity most relevant to hierarchical refinement.
>
> Although any segmentation strategy introduces the potential for structural discontinuities, our analyses demonstrate that **PAE-guided decomposition is robust**, **does not introduce harmful structural breaks**, and **is further safeguarded by interface-aware selection and full refinement in the Fine Stage**. The method performs consistently across a wide range of threshold choices, and empirical comparisons confirm that PAE remains a reliable structural signal for hierarchical complex prediction.
>
> We further compared our PAE-based segmentation method with other subunit/domain segmentation methods. The PAE-based segmentation method **outperforms other methods on complex structure prediction**. Please refer to the **common concern 1** for details.

---

> ### Author Response · Authors · 2025-11-23
>
> ### **Question 3: Assembly Mechanism for Flexibility and Structural Strain**
>
> We thank the reviewer for raising this important question regarding rigid-body versus flexible motions during assembly. While the assembly stage performs only **global rigid alignment**, our hierarchical design explicitly avoids introducing structural strain and is robust to flexibility for the following reasons.
>
> **1. No structural concatenation across independently predicted subunits**
>
> In each Fine Stage round, the predictor performs a *full AlphaFold3 inference* on a **single focus chain**, while all other chains are provided only as structural context. After refinement, **only the updated coordinates of the focus chain are retained**. The context chains are *not* stitched or merged across rounds.
>
> This design ensures that **no two independently predicted subunits are ever directly concatenated**, preventing discontinuities or strain accumulation at subunit boundaries.
>
> **2. Rigid-body alignment applies only to globally consistent subunit poses**
>
> During assembly, we compute a transformation matrix that aligns the newly refined focus-chain pose with the existing partial complex using the predicted inter-chain frames. Because only the focus chain’s conformation is being updated—and the rest serve merely as reference—any structural adjustments are absorbed **within the focus chain**, not distributed across multiple subunits.
>
> Thus, rigid alignment handles **global pose differences**, while **local backbone and side-chain rearrangements** are fully resolved by AlphaFold3 during the Fine Stage.
>
> **3. Flexible regions naturally receive lower influence during assembly**
>
> Flexible or poorly predicted segments typically carry **low confidence (pLDDT/PAE)**. Our confidence-weighted assembly mechanism down-weights these regions when computing the alignment transformation. This prevents noisy or flexible residues from exerting undue influence on the global alignment, thereby reducing strain and avoiding the propagation of errors from unstructured regions.
>
> **4. The Fine Stage inherently captures local flexibility**
>
> Because each focus-chain pass is a complete AlphaFold3-style inference:
>
> - local backbone corrections,
> - hinge movements,
> - helix/loop rearrangements, and
> - side-chain repacking
>
> are all handled by the predictor itself. The assembly step does not override or distort these adjustments; it only positions the refined subunit coherently within the complex.
>
> Although the assembly step is implemented as a rigid-body alignment, the overall workflow **preserves structural integrity** because:
>
> 1. **Only the focus chain is updated and retained**, eliminating strain from merging independently predicted subunits.
> 2. **Global rigid transformations** handle inter-subunit poses, while **local flexibility** is fully modeled during the Fine Stage.
> 3. **Confidence weighting** prevents flexible regions from introducing instability.
>
> Together, these mechanisms ensure that the hierarchical assembly does not introduce structural strain and can robustly handle both rigid and flexible components of large complexes.
>
> To address the reviewers' concerns, we have conducted several additional experiments and analyses, including:
>
> - A comprehensive sensitivity analysis of the PAE split threshold to validate decomposition robustness.
> - Validation of the pipeline’s robustness against Intrinsically Disordered Regions (IDRs) as a proxy for structural ambiguity.
> - Integration of RoseTTAFold All-Atom to demonstrate the framework's modularity and compatibility with future predictors.
> - Clarification of the assembly mechanism regarding structural strain and flexibility.
>
> All revisions in the paper are marked in blue font. We sincerely appreciate the reviewers' constructive suggestions and remain committed to further improving our work.

---

### Official Review · Reviewer_sJfG · 2025-11-01

**Soundness:** 3
**Presentation:** 2
**Contribution:** 2
**Rating:** 6
**Confidence:** 3

**Summary:**

The paper introduces a new approach for predicting the structure of large protein complexes by decomposing them into modular subunits using Predicted Aligned Error (PAE) scores. The proposed approach uses a 3-stage process from coarse to split the segments, then performing fine prediction using existing models and the final alignment. It achieves similar accuracy to AlphaFold3 with lower GPU memory requirements for large protein complex.

**Strengths:**

- The paper addresses a difficult challenge of predicting the structure of large protein complexes.

- The proposed approach appears to improve both accuracy and reduce memory requirements.

- The three stages of coarse, fine, and assembly are modular and allow for examining individual stages. Overall, the approach is well constructed.

**Weaknesses:**

- The proposed approaches rely on existing approaches in protein modeling and do not advance the area significantly, nevertheless, they provide an application to an important problem.

  - The reliance of PAE for segmentation is one of the main drawbacks of this approach. PAE is an estimated alignment error and may not fully capture the inter-domain interactions, leading to errors in clustering based only on PAE.

  - Since the main comparison in this paper was done between hierafold and combfold, it would be helpful to compare against the dataset used in the combfold paper.

  - Description of the dataset and performance variation with respect to homology, number of conformations, or known multi-state assemblies is missing. The paper would be strengthened if it explicitly demonstrated performance on disordered/flexible domains.

  - A better quantification or analysis of the identified sub-unit from segmentation is not provided. This will also allow to examine if common sub-structures are being identified as subunits across different proteins.

**Questions:**

Details of the dataset are very sparse. Provide more details, including the number and lengths of sequences across the 3 datasets, in the manuscript.

---

> ### Author Response · Authors · 2025-11-23
>
> Dear Reviewer sJfG,
>
> We sincerely thank the reviewer for the constructive feedback and for recognizing our work. Below, we address the specific weaknesses and questions raised, with new data and analyses added to the revised manuscript.
>
> ### **Weakness 1: The reliance of PAE for segmentation is one of the main drawbacks of this approach**
>
> We thank the reviewer for raising this important point. PAE indeed estimates the expected alignment error between residue pairs, and therefore encodes the model’s belief about the *relative flexibility and coherence* of structural regions. When residues belong to the same rigid unit or interact in a stable manner, their relative positions are predicted with higher confidence—manifesting as **low-PAE blocks**. Conversely, flexible linkers, disordered regions, or weakly interacting segments naturally exhibit **high PAE**, providing a meaningful structural signal for segmentation.
>
> Our PAE-guided segmentation is intentionally designed as a **lightweight, inference-time method** to extract *structurally rigid subunits* that are most relevant to hierarchical refinement. Unlike evolutionary domain parsers, which classify domains based on sequence homology, our method identifies **functional, rigidity-based partitions** that directly align with the requirements of downstream structure prediction. For interface detection, we further combine PAE with **predicted inter-residue distances**, which helps robustly capture interacting subunit pairs even when PAE alone may be ambiguous.
>
> To address the reviewer’s concern empirically, we newly conducted experiments comparing our PAE-based segmentation with **Merizo**, a state-of-the-art deep learning–based domain segmentation model. **Please refer to the common concern 1 for details and more discussion of PAE-based segmentation.** The results (summarized below) demonstrate that PAE is an effective and reliable signal for the segmentation *needed in HierAFold*, even if it is not optimized for canonical domain annotation.
>
> 1. **Segmentation accuracy on CATH-663**
>     - Merizo achieves a higher IoU (0.85) than our method (0.65).
>
>         This is expected because Merizo is explicitly trained for domain parsing, including discontinuous domains.
>
>     - Our method achieves **boundary MCC = 0.79**, close to Merizo’s 0.84.
>
>         This indicates that **PAE successfully identifies the true structural boundaries**, even if it produces coarser segmentation.
>
> 2. **Downstream structure prediction performance**
>     - When replacing PAE-based segmentation with Merizo in HierAFold, **performance decreases**:
>         - **Merizo + HierAFold:** 72.6% Oracle DockQ / 68.4% Top-1
>         - **PAE-based HierAFold:** **73.1% Oracle / 69.0% Top-1**
>     - This shows that **PAE-based segmentation is better aligned with hierarchical refinement**, despite being simpler and computationally cheaper.
> 3. **Practical efficiency**
>     - PAE segmentation is **computationally free**, reusing confidence maps already produced during coarse prediction.
>     - In contrast, Merizo requires an additional neural network forward pass.
>
> Together, these results validate that **PAE provides a sufficient and robust structural signal** for identifying the modular subunits needed by HierAFold. While PAE may not recover fine-grained evolutionary domain boundaries, it effectively captures **rigidity, interaction coherence, and boundary transitions**, enabling accurate and scalable hierarchical refinement.
>
> For a more detailed discussion, please refer to our response to **Common Concern 1** and the updated experimental section in the revised manuscript.
>
> ### **Weakness 2: It would be helpful to compare against the dataset used in the CombFold paper.**
>
> Thanks for the suggestions. We have added Benchmark 2 in the CombFold paper, since the release date of some cases in Benchmark 1 and 3 is before the AlphaFold3 training date cut-off. The DockQ success rate (**30.2% (CombFold) vs 50.7% (ours)**) is reported in Appendix A.5 in the revised paper.

---

> ### Author Response · Authors · 2025-11-23
>
> ### **Weakness 3: Description of the dataset and performance variation with respect to homology, number of conformations, or known multi-state assemblies is missing. The paper would be strengthened if it explicitly demonstrated performance on disordered/flexible domains.**
>
> Thank you for pointing out all these missing details. All details are included in the revised manuscript.
>
> ### **3.1 Description of the dataset**
>
> We have expanded the dataset descriptions to provide clearer context for our benchmarks. Specifically:
>
> - **Recent PDB dataset:** Contains complexes with up to **20 chains** and **5,120 tokens**, following the definitions used in Protenix and AlphaFold3.
> - **Filtered Large Complex dataset:** Includes complexes with up to **30 chains** and **10,000 tokens**, designed to evaluate scalability beyond the range handled by AlphaFold3.
> - **PoseBuster v2:** Comprises protein–ligand assemblies with up to **4,100 tokens** and **4 chains**, serving as a test for ligand docking performance.
>
> These details are now included in the dataset section of the revised manuscript.
>
> ### **3.2 Description of the homology**
>
> We have added a new analysis in Appendix A.8 examining performance across **homomeric** versus **heteromeric** complexes. Complexes in the recent PDB dataset were categorized based on chain composition. As shown in Table 1, both AlphaFold3 and **HierAFold** achieve higher accuracy on homomers, likely due to stronger co-evolutionary and symmetry-related signals:
>
> | **Category** | **Metric** | **AlphaFold3** | **HierAFold** |
> | --- | --- | --- | --- |
> | **Homomers** | Oracle | 77.3 | 76.2 |
> |  | Top-1 | 74.0 | 72.8 |
> | **Heteromers** | Oracle | 72.3 | 70.9 |
> |  | Top-1 | 67.9 | 66.3 |
>
> This comparison is now included in the manuscript to clarify performance variation with homology.
>
> ### **3.3 Description of the number of conformations**
>
> We have added the discussion about the number of conformations in Appendix A.9 in the revised manuscript.
>
> Our experiments in recent PDB follow the standard inference protocol of Protenix, generating 25 conformations. To assess the impact of sampling, we evaluated performance when generating fewer conformations. The results are summarized in Table 2.
>
> | **Category** | **Metric** | **AlphaFold3** | **HierAFold (Ours)** |
> | --- | --- | --- | --- |
> | **n=5** | Oracle | 69.9 | 68.5 |
> |  | Top-1 | 65.0 | 63.6 |
> | **n=10** | Oracle | 72.2 | 70.9 |
> |  | Top-1 | 67.5 | 66.1 |
> | **n=25** | Oracle | 74.4 | 73.1 |
> |  | Top-1 | 70.4 | 69.0 |
>
> ### **3.4 Description of known multi-state assemblies**
>
> We have added the discussion about known multi-state assemblies in Appendix A.12.2 in the revised manuscript.
>
> A known challenge for AlphaFold-family models is the prediction of distinct functional states. Since **HierAFold** uses AlphaFold3 as its predictor, it inherits this limitation. For example, when predicting the structures of an E3 ubiquitin ligase in its apo (PDB: 8cvp, expected open state) and holo (PDB: 8d7u, expected closed state) forms, **HierAFold**, like the baseline, exclusively predicts the closed conformation for both (Fig. 9 in revised manuscript). Methods developed for AlphaFold-family models that assist in multistate prediction may be adaptable in **HierAFold**.
>
> ### **3.5 Description of performance on disordered/flexible domains**
>
> A detailed evaluation of **HierAFold** on intrinsically disordered regions (IDRs) is presented in our response to **Common Concern 2** and is now fully integrated into the revised manuscript. This analysis demonstrates that **HierAFold** maintains performance comparable to AlphaFold3 across varying levels of disorder while enabling prediction far beyond the scale at which AlphaFold3 can operate.
>
> ### **Weakness 4: A better quantification or analysis of the identified sub-unit from segmentation is not provided.**
>
> We address this point thoroughly in our responses to **Weakness 1** and **Common Concern 1**, where we present quantitative analyses of the segmentation quality, comparisons against established methods (e.g., Merizo), and evaluation of how segmentation impacts downstream prediction accuracy.
>
> ### **Question 1: Details of the dataset are very sparse.**
>
> Please refer to Weakness 3.1. We have added the details of the dataset in the revised paper.
>
> To address the reviewers' concerns, we have conducted several additional experiments and analyses, including:
>
> - A comprehensive analysis of our PAE-based segmentation method and comparison with other segmentation methods.
> - Additional benchmarking on the CombFold dataset.
> - Detailed performance breakdowns regarding homology (homomers vs. heteromers), number of generations, and multi-state assemblies.
> - Validation of the pipeline’s robustness against Intrinsically Disordered Regions (IDRs).
>
> All revisions in the paper are marked in blue font. We sincerely appreciate the reviewers' constructive suggestions and remain committed to further improving our work.

---

### Official Review · Reviewer_eHdb · 2025-11-01

**Soundness:** 4
**Presentation:** 4
**Contribution:** 4
**Rating:** 6
**Confidence:** 3

**Summary:**

The method HIERAFOLD provides a memory-efficient and accurate solution to the prediction of large macromolecular complexes. It uses an optimised version of Protenix, an open-source reproduction of AlphaFold3, to generates 3D models of subparts, a subunit segmentation strategy based on PAE, and a combinatorial algorithm for selecting and assembling subparts. It performs favourably compared to AlphaFold3 and CombFold.

**Strengths:**

- The methodology and experiments are sound and clearly explained
- The success rate is much better than the state-of-the-art methods

**Weaknesses:**

- The main concern I have is whether this work actually fits within the scope of ICLR. It is not a machine learning contribution per se, as it relies on an already developed diffusion-based SOTA protein structure predictor. The contribution mainly focuses on algorithms for segmenting, selecting and combining subparts, without any representation learning involved.

**Questions:**

- Could the authors discuss other methods for segmenting proteins into domains, or rigid subunits? In particular those developed recently for annotating the AlphaFold Database?
- Could the authors comment on the influence of the content in intrinsically disordered regions on the performance of the method?

---

> ### Author Response · Authors · 2025-11-23
>
> Dear Reviewer eHdb,
>
> We greatly appreciate your recognition of **HierAFold**. We are also grateful for your constructive comments and questions. We have carefully addressed your questions regarding protein segmentation methods and the influence of Intrinsically Disordered Regions (IDRs) in the revised manuscript. Below is a detailed response to your comments.
>
> ### **Weakness 1: Scope of the work within ICLR**
>
> We appreciate the reviewer’s concern and welcome the opportunity to clarify why **HierAFold** fits squarely within the scope and interests of ICLR. Although our method builds on an existing diffusion-based structure predictor, our contributions are fundamentally rooted in **representation understanding**, **algorithmic innovation leveraging learned confidence signals**, and **learning-enhanced hierarchical inference**, all of which align well with ICLR’s focus.
>
> ### **1. Learning-Derived Representations as the Core Algorithmic Signal**
>
> A central novelty of **HierAFold** is its use of the **Predicted Aligned Error (PAE)**—a rich, *learned representation* produced by the underlying model. Rather than treating PAE as a post-hoc diagnostic, we reinterpret it as a **machine-learned structural prior** that encodes rigidity, relational uncertainty, and cross-chain interaction patterns. Our algorithmic framework is built around *understanding* and *exploiting* this learned representation to:
>
> - segment chains into **rigid structural subunits**,
> - identify **cross-chain interface subunits**, and
> - construct a **hierarchical dependency graph** for refinement.
>
> This constitutes a novel use of model-internal learned signals for decomposition and inference—a direction increasingly emphasized in ICLR (e.g., interpretability, using internal representations for decision-making, leveraging confidence fields for structure).
>
> ### **2. Consistency-Distilled Diffusion Model**
>
> To support scalable hierarchical inference, we introduce a **new, lightweight diffusion model** trained through **consistency distillation**. Specifically:
>
> - we design a **shallower diffusion architecture** for pairwise coarse prediction,
> - train it using a **consistency-based objective** (Eq. 1), and
> - achieve **few-step inference** while preserving distributional fidelity.
>
> This distilled model is not merely an engineering component—it is a *learned* approximation of the full diffusion predictor tailored for hierarchical modular inference.
>
> ### **3. Alignment with ICLR Subject Areas**
>
> The ICLR call for papers explicitly includes several categories that match our contributions:
>
> - **Applications to the physical sciences (physics, chemistry, biology)**
>
>     Our work applies modern ML methodology to a major open challenge in structural biology—scaling deep generative models to large complexes.
>
> - **Structured prediction**
>
>     Protein complex modeling is a canonical structured prediction problem involving geometric, relational, and hierarchical structure. We introduce a **new hierarchical structured prediction framework** that operates over learned relational fields (PAE).
>
> - **Efficient inference / scalable ML**
>
>     Our hierarchical refinement pipeline addresses the *scalability bottleneck* of diffusion models, achieving up to 40% memory saving and enabling prediction beyond the token limit of state-of-the-art models.
>
>
> In summary, **HierAFold** is not simply an engineering wrapper around AlphaFold3. It proposes:
>
> 1. **A machine-learning–driven hierarchical inference framework**,
> 2. **A novel algorithmic use of learned representations (PAE) for modular decomposition**, and
> 3. **A distilled diffusion model enabling scalable structured prediction**.
>
> We believe these contributions fit naturally within the scope of ICLR and provide value to both the ML community and scientific applications.
>
> ### **Q1: Other methods for segmenting proteins into domains, or rigid subunits?**
>
> Thank you for the suggestion. Please refer to the **common concern 1** for a detailed discussion.
>
> ### **Q2: Comment on the influence of the content in intrinsically disordered regions on the performance of the method**
>
> We thank the reviewer for raising this important point. Please refer to the **common concern 2** for detailed performance on IDRs.
>
> To address the reviewers' concerns, we have conducted several additional experiments and analyses, including:
>
> - A comprehensive comparison and analysis of other protein domain/subunit segmentation methods
> - Validation of the pipeline’s robustness against Intrinsically Disordered Regions (IDRs).
>
> All revisions in the paper are marked in blue font. We sincerely appreciate the reviewers' constructive suggestions and remain committed to further improving our work.

---

> > ### Comment · Reviewer_eHdb · 2025-11-27
> >
> > I would like to thank the authors for addressing my concerns. In the meantime, I noticed the code is not publicly available. I would like to keep my score.

---

> > > ### Author Response · Authors · 2025-11-27
> > >
> > > Dear reviewer eHdb,
> > >
> > > Thank you once again for taking the time to review our work and for acknowledging it. We would like to assure you that the complete code, including both the training and inference stages, will be made publicly available upon the acceptance of the paper. In the meantime, if you have any questions or require clarification regarding our method or implementation details, we would be happy to address them.

---

### Author Response · Authors · 2025-11-23
**Common Concern2**

### **Concern 2: Performance on Intrinsically Disordered Regions (IDRs)**

Intrinsically disordered protein regions (IDRs), sequences lacking stable tertiary structure, are pivotal in cellular processes. Interactions between IDRs and their partners are characterized by partial and dynamic binding and are accurately captured by AFM.

To evaluate how IDRs affect **HierAFold**, we analyzed the recent PDB dataset following established procedures. Each residue was annotated with a disorder probability using AIUPred, interface residues were defined as heavy-atom contacts within 10 Å of each other, and for each complex we computed the **mean disorder score at the interface**. Complexes were then grouped into three bins: low (0.1–0.25), medium (0.25–0.5), and high disorder (> 0.5). For both the AlphaFold3 baseline and **HierAFold**, we measured the mean DockQ for each bin:

| **Mean Interface IDR Score** | **AlphaFold3 (Baseline)** | **HierAFold (Ours)** |
| --- | --- | --- |
| Low (0.1–0.25) | 0.55 | 0.52 |
| Medium (0.25–0.5) | 0.49 | 0.44 |
| High (> 0.5) | 0.46 | 0.41 |

The results reveal a consistent trend: as interface disorder increases, DockQ scores decrease for both models. This is expected, because IDRs are flexible, underrepresented in training data, and often participate in transient or fuzzy interactions.

Importantly, **HierAFold exhibits almost the same degradation pattern as AlphaFold3**, demonstrating that our hierarchical coarse-to-fine pipeline preserves AlphaFold3’s robustness when handling disordered interfaces. In other words, **the introduction of subunit segmentation, interface selection, and hierarchical assembly does not introduce additional weaknesses for IDRs**.

This robustness stems from how IDRs behave under our PAE-guided workflow:

- **Segmentation:** IDRs appear as high-PAE or highly flexible regions and are naturally isolated into their own subunits.
- **Interaction Selection:** When IDRs genuinely participate in binding, they produce localized low cross-chain PAE signals that ensure their inclusion during fine-stage refinement.
- **Assembly:** Because IDRs generally receive low confidence scores, they are automatically down-weighted during confidence-weighted alignment, preventing unreliable geometry from distorting the global structure.

Overall, **HierAFold matches AlphaFold3’s ability to model disordered interfaces**, while simultaneously overcoming the memory limitation that prevents AlphaFold3 from being applied to large complexes, where IDRs are even more prevalent. Thus, our results show that **PAE-guided hierarchical modeling maintains IDR robustness while extending applicability far beyond the scale accessible to AlphaFold3**.

---

### Author Response · Authors · 2025-11-23
**Common Concerns**

Dear all reviewers,

We thank reviewers for all their feedback, which we believe will help strengthen the **HierAFold**. Below, we address two common concerns and provide updates and discussions about them. The first concern is about comparing with other subunit/domain segmentation methods, and the second is about performance on Intrinsically Disordered Regions (IDRs).

### **Q1: Comparison with other protein subunit/domain segmentation methods**

We thank the reviewer for raising the important point regarding comparisons with existing protein subunit and domain segmentation methods. In the revised manuscript, we have significantly expanded the **Related Work section** to include a detailed discussion of domain/subunit segmentation approaches—both database-driven and learning-based—and we provide a direct comparison to our PAE-based strategy. We have also added new experiments validating our design choice.

### **1. Comparison with Existing Subunit/Domain Segmentation Methods**

Protein domain segmentation has been studied, with two primary categories of methods:

- **Domain annotation databases**, such as CATH and ECOD, which provide curated evolutionary classifications and domain boundaries based on sequence and structure similarity.
- **Computational prediction methods**, including: Sequence-based predictors, Structure-based segmentation algorithms, Deep-learning approaches such as Merizo and Chainsaw.
- **Large-scale efforts**, such as TED, which annotate structural domains across the protein universe by integrating these advanced tools.

These approaches are highly effective for identifying evolutionarily conserved domains, but they also carry practical limitations:

- They often require additional inference-time computation, external databases, or heavyweight neural models.
- They are optimized for evolutionary domain definitions, rather than the structurally rigid units relevant to downstream complex modeling.
- They do not identify cross-chain interface subunits, as domain databases annotate individual polypeptide chains only.

In contrast, our method uses **PAE matrices already produced during HierAFold’s coarse prediction stage**, and thus introduces **no additional computational overhead**. The PAE signal naturally reflects structural rigidity and interface confidence, enabling us to segment chains into **intrinsically rigid subunits** and **cross-chain interface subunits**, which existing domain segmentation tools cannot provide. This makes PAE-based segmentation more directly aligned with the goals of complex assembly and multi-chain refinement in HierAFold.

### **2. Experimental Validation with Merizo**

To directly evaluate whether more sophisticated domain segmentation methods improve downstream complex prediction, we replaced our PAE-based module with Merizo, a state-of-the-art deep-learning domain segmentation model. We created a Merizo + HierAFold variant while keeping all other components unchanged. Results on the recent PDB benchmark (Table below) show that Merizo-based segmentation leads to lower overall prediction accuracy, while also incurring substantially higher additional computational and resource costs for segmentation.

| **Method** | **Oracle DockQ (%)** | **Top-1 DockQ (%)** | **Segmentation Time (3000 tokens)** |
| --- | --- | --- | --- |
| Merizo + **HierAFold** | 72.6 | 68.4 | 50s |
| **HierAFold** | **73.1** | **69.0** | 1s |

This experiment demonstrates that more complex domain segmentation does not yield better downstream structure prediction and confirms that PAE-based segmentation is well aligned with HierAFold’s refinement pipeline without additional costs.

---

> ### Author Response · Authors · 2025-11-23
> **Common Concern1 Part2**
>
> ### **3. Segmentation Accuracy Comparison**
>
> To contextualize our approach, we compared the segmentation accuracy of our PAE-guided method with Merizo on the CATH-663 benchmark. As expected, Merizo achieves higher IoU (0.85 vs. 0.65), since it is explicitly designed and trained for **domain parsing**, including the detection of **discontinuous domains** that are prevalent in CATH. In contrast, our segmentation module is not optimized for canonical domain detection, yet still achieves reasonable segmentation quality. It deliberately merges tightly packed regions with uniformly low inter-subunit PAE into a single rigid unit—a behavior aligned with our design objective of identifying **structurally coherent subunits** that matter for downstream refinement, rather than reproducing evolutionary domain boundaries.
>
> Despite this difference in segmentation granularity, our method achieves a **boundary MCC of 0.79**, closely matching Merizo’s 0.84. This demonstrates that **PAE provides a reliable structural signal for locating true physical boundaries**, even without specializing in fine-grained domain delineation.
>
> Crucially, these variations in IoU have **no negative impact on the final complex prediction accuracy**. The results confirm that **PAE is an effective and sufficient metric for the type of segmentation required by HierAFold**—identifying rigid subunits and interaction-relevant boundaries that guide accurate multi-chain refinement.
>
> ### **4. Why PAE-Based Segmentation is Preferable for HierAFold?**
>
> Combined with the results above, our PAE-based segmentation offers three key advantages:
>
> - **Computational Efficiency**: No additional model or database queries are required. Segmentation reuses PAE matrices already produced during coarse prediction, making it effectively cost-free.
> - **Structural Relevance for Complex Assembly**: PAE identifies rigid, cooperatively moving structural units, not evolutionary domains. It also highlights inter-chain interaction subunits, enabling interface-aware refinement—capabilities absent in existing domain predictors.
> - **Competitive or Better Downstream Performance**: Despite simpler computation, PAE-based segmentation performs better than Merizo when evaluated in the context of complex structure prediction, the task for which HierAFold is optimized.
>
> In summary, while existing domain segmentation tools are powerful and well-established, they are not tailored for the structural modularity and interface-focused refinement required in large complex prediction. Our PAE-based approach is: aligned with the model’s confidence signal, computationally free, capable of capturing cross-chain interface subunits, and empirically superior for HierAFold’s downstream prediction accuracy. These additions and comparisons are now fully incorporated into the revised manuscript.

---

### Author Response · Authors · 2025-12-03
**Final Remarks and Summary to AC**

Dear AC,

We thank you and all reviewers for the time spent reviewing our paper. We summarize the key points below.

| **Reviewer** | **Strengths Highlighted** | **Key Questions / Concerns** | **Initial Rating** | **Our Reply & New Experiments** | **Final Feedback / Status** |
| --- | --- | --- | --- | --- | --- |
| **eHdb** | • Sound methodology • Excellent success rate vs SOTA • Clear presentation | • **Scope:** Whether this work fit ICLR • **Segmentation:** Compare with other methods (e.g., Merizo). • **IDRs:** Influence of disordered regions. | **6** | • Clarified ML contributions (distilled diffusion, learned PAE representations). • **New Exp:** Compared with Merizo (PAE is cost-free & accurate for this task). • **New Exp:** IDR analysis showing robustness. | Concerns resolved. Raised concern about code availability (we confirmed code will be public). |
| **sJfG** | • Addresses difficult challenge • Modular design • Improves accuracy & memory | • **PAE Reliance:** Concerns about segmentation accuracy. • **Datasets:** Missing details and variance on homology and conformation. & CombFold comparison. • **IDRs:** Performance on flexible domains. | **6** | • **New Exp:** Compared with Merizo. • **New Exp:** Added CombFold benchmark 2. • **New Exp:** Homology & multi-state & number of conformations analysis. • **New Exp:** IDR robustness validation. • Clarified dataset specs. | *(No post-rebuttal response)* |
| **EYwv** | • High originality (PAE decomposition) • Biologically motivated • Significant practical value | • **Robustness:** Sensitivity to initial subunit quality. • **Structural Breaks:** Risk of artifacts at boundaries. • **Assembly:** Handling flexibility. | **4** | • **New Exp:** Sensitivity analysis of PAE threshold (\tau_{split}) showing stability. • Explained Fine Stage correction mechanism. • Clarified confidence-weighted assembly. | *(No post-rebuttal response)* |
| **E9v3** | • Clear motivation • Effective experiments | • **Backbones:** Only used AlphaFold3. • **Baselines:** Need OpenFold/MoLPC. • **Cost:** Computation vs. Memory trade-off. • **Code:** Not public. • **Sensitivity**: PAE segmentation sensitivity. | **4** | • **New Exp:** Integrated **RoseTTAFold All-Atom** (showed generalizability). • **New Baseline:** RoseTTAFold All-Atom as baseline • **New Analysis:** Detailed computational cost & mitigation strategies. • Confirmed code release. • **New Exp**: Sensitivity analysis of PAE threshold (\tau_{split}) showing stability. | Acknowledged concerns were addressed. |
| **XQxJ** | • Important problem (scalability) • No expert curation needed • Substantial improvements | • **Writing:** Organization/Typos. • **Cost:** Overhead analysis. • **Design:** Validate PAE vs. other metrics. • **Intuition:** Why PAE? | **4** | • **New Exp:** Ablation of PAE vs. PDE (PAE superior). • **New Exp:** Ablation of Fine Stage necessity. • Revised writing & added Related Works section. • Analyzed performance gap vs. size. | Concerns resolved. Score raised to 6.  |

---

### **Key Revisions & New Experiments Summary**

To address the common concerns raised across multiple reviewers, we have incorporated the following major revisions and new experiments into the manuscript:

1. **Comparison with Domain Segmentation Methods (Merizo):**

    We added a thorough comparison against Merizo, a state-of-the-art protein domain segmentation model. While Merizo achieves higher IoU for evolutionary domain labels, our **PAE-based segmentation** achieves comparable boundary accuracy (MCC 0.79 vs. 0.84), delivers **higher downstream structure prediction accuracy** (DockQ 69.0% vs. 68.4%), and incurs **no additional computational cost**. (Response to eHdb, sJfG, XQxJ).

2. **Robustness on Intrinsically Disordered Regions (IDRs):**

    We evaluated performance across low-, medium-, and high-disorder interfaces. The results show that **HierAFold preserves AlphaFold3-level robustness on IDRs** while enabling inference on large complexes that AlphaFold3 cannot process due to memory limits. Response to eHdb, sJfG, EYwv).

3. **Generalizability Across Backbone Models:**

    We integrated **RoseTTAFold All-Atom** into the HierAFold pipeline. The hierarchical version achieves accuracy comparable to the end-to-end model while reducing memory consumption by ~35%, demonstrating that **HierAFold is model-agnostic and easily extendable to future predictors**.  (Response to E9v3, EYwv)

4. **Sensitivity & Computational Cost Analysis:**

    We added ablations on the PAE segmentation threshold, showing that performance remains stable across a wide range of values. In addition, we provide a detailed analysis of **runtime vs. memory savings**, highlighting that the modest computational overhead is outweighed by the ability to handle large complexes that cause OOM errors in baseline models. (Response to EYwv, E9v3, XQxJ).

---

### Meta-Review · Area_Chair_qDwp · 2026-01-06

**Summary:**

This study presents HierAFold, a hierarchical pipeline designed to address the quadratic memory growth bottleneck of state-of-the-art protein structure predictors (e.g., AlphaFold3) for large complexes (> a few thousand tokens). It leverages PAE-guided subunit decomposition, targeted interface-aware refinement, and confidence-weighted assembly to exploit the modularity of large complexes. Strong empirical performance was reported. HierAFold matches AlphaFold3’s accuracy, significantly raises success rates on recent PDB sets, cuts peak memory by ~40% and successfully models complexes with over 5,000 tokens that are out-of-memory for AlphaFold3.

All the reviewers agree the study focuses on a critical problem and provides a creative solution. Strong performance is achieved, especially the method successfully models complexes with over 5,000 tokens that are out-of-memory for AlphaFold3 (raised by Reviewer eHdb, sJfG, E9v3, XQxJ).

The key concerns from reviewers are:

**1. This study shows limited methodological contribution and scope (raised by Reviewer eHdb, sJfG, EYwv)**

The approach relies on existing protein modeling techniques and does not advance the field significantly, despite its practical utility.

**2. Potential limitations of PAE-guided decomposition approach (raised by Reviewer sJfG, EYwv, E9v3 and XQxj)**

Reviewer sJfG and XQxJ: The rationale for using low-RAE regions to identify critical subunits is not explained. PAE is an estimated alignment error and may not fully capture inter-domain interactions, leading to clustering errors.

Reviewer sJfG, E9v3 and XQxJ: Sensitivity analysis should be provided to show how PAE decomposition threshold affects the performance.

**3. Insufficient evaluation (raised by Reviewer sJfG, E9v3, XQxJ)**

Reviewer sJfG and E9v3: Comprehensive evaluations using more SOTA models (OpenFold, CombFold) and divide-and-conquer approaches should be provided.

Reviewer E9v3 and EYwv: Generalizability across structure prediction models

Reviewer EYwv: Sensitivity of the method to initial subunit prediction quality.

Reviewer EYwv, XQxJ: Can the confidence weighted assembly handle the difference between rigid-body movements of subunits (which require global transformation) and local, flexible changes?

Reviewer XQxJ: Limited ablation studies to validate the design choices of the method.

**Reviewer Concerns:**

I believe the authors have successfully addressed the majority of the key concerns raised by the reviewers.

**1. This study shows limited methodological contribution and scope (raised by Reviewer eHdb, sJfG, EYwv)**
The authors have clearly clarified the dual contributions of their work, providing explicit and detailed justifications for its relevance to both the machine learning field and the broader scientific domain of structural biology.

**2. Potential limitations of PAE-guided decomposition approach (raised by Reviewer sJfG, EYwv, E9v3 and XQxj)**
This critical concern has been fully addressed by the authors, who supplemented the manuscript with additional experimental results. These new analyses demonstrate that PAE can reliably and accurately decompose protein structures into functional subunits, with the subsequent downstream structure prediction retaining strong and consistent performance.

**3. Insufficient evaluation (raised by Reviewer sJfG, E9v3, XQxJ)**
The authors have thoroughly resolved this concern by presenting a comprehensive suite of new experiments.

**Reviewer Scores:**

My anticipated revised scores for each reviewer are as follows:

Reviewer eHdb: Maintain the original score of 6.

Reviewer sJfG: Key concerns have been fully addressed; the score may be increased to 6.

Reviewer EYwv: Potential score increase from 4 to 6 (uncertain).

Reviewer E9v3: Key concerns have been fully addressed; the score may be increased to 6.

Reviewer XQxJ: Key concerns have been fully addressed; the score may be raised to 6.

---

### Decision · Program_Chairs · 2026-01-26

Accept (Poster)